# pcaGAN: Improving Posterior-Sampling cGANs via Principal Component Regularization

**Matthew C. Bendel**
Dept. ECE
The Ohio State University
Columbus, OH 43210
`bendel.8@osu.edu`

**Rizwan Ahmad**
Dept. BME
The Ohio State University
Columbus, OH 43210
`ahmad.46@osu.edu`

**Philip Schniter**
Dept. ECE
The Ohio State University
Columbus, OH 43210
`schniter.1@osu.edu`

## Abstract

In ill-posed imaging inverse problems, there can exist many hypotheses that fit both the observed measurements and prior knowledge of the true image. Rather than returning just one hypothesis of that image, posterior samplers aim to explore the full solution space by generating many probable hypotheses, which can later be used to quantify uncertainty or construct recoveries that appropriately navigate the perception/distortion trade-off. In this work, we propose a fast and accurate posterior-sampling conditional generative adversarial network (cGAN) that, through a novel form of regularization, aims for correctness in the posterior mean as well as the trace and K principal components of the posterior covariance matrix. Numerical experiments demonstrate that our method outperforms contemporary cGANs and diffusion models in imaging inverse problems like denoising, large-scale inpainting, and accelerated MRI recovery. The code for our model can be found here: `https://github.com/matt-bendel/pcaGAN`.

## 1   Introduction

In image recovery, the goal is to recover the true image $x$ from noisy/distorted/incomplete measurements $y = \mathcal{M}(x)$. This arises in linear inverse problems such as denoising, deblurring, inpainting, and magnetic resonance imaging (MRI), as well as in non-linear inverse problems like phase-retrieval and image-to-image translation. For all such problems, it is impossible to perfectly recover $x$ from $y$.

In much of the literature, image recovery is posed as point estimation, where the goal is to return a single best estimate $\widehat{x}$. However, there are several shortcomings of this approach. First, it's not clear how to define "best," since L2- or L1-minimizing $\widehat{x}$ are often regarded as too blurry, while efforts to make $\widehat{x}$ perceptually pleasing can sacrifice agreement with the true image $x$ and cause hallucinations [1–5], as we show later. Also, many applications demand not only a recovery $\widehat{x}$ but also some quantification of uncertainty in that recovery [6, 7].

As an alternative to point estimation, *posterior-sampling*-based image recovery [8–27] aims to generate $P \geq 1$ samples $\{\widehat{x}_i\}_{i=1}^P$ from the posterior distribution $p_{\mathsf{x}|\mathsf{y}}(\cdot|y)$. Posterior sampling facilitates numerous strategies to quantify the uncertainty in estimating $x$, or any function of $x$, from $y$ [6, 7]. It also can help with visualizing uncertainty and increasing robustness to adversarial attacks [28]. That said, the design of accurate and computationally-efficient posterior samplers remains an open problem.

Given a training dataset of image/measurement pairs $\{(x_t, y_t)\}_{t=1}^T$, our goal is to build a generator $G_{\boldsymbol{\theta}}$ that, for a given $y$, maps random code vectors $z \sim \mathcal{N}(0, I)$ to samples of the posterior, i.e., $\widehat{x} = G_{\boldsymbol{\theta}}(z, y) \sim p_{\mathsf{x}|\mathsf{y}}(\cdot, y)$. There are many ways to approximate the ideal generator. The recent literature has focused on conditional variational autoencoders (cVAEs) [12–14], conditional

normalizing flows (cNFs) [8–11], conditional generative adversarial networks (cGANs) [15–19], and Langevin/score/diffusion-based generative models [20–27]. Although diffusion models have garnered a great share of recent attention, they tend to generate samples several orders-of-magnitude slower than their cNF, cVAE, and cGAN counterparts.

With the goal of fast and accurate posterior sampling, recent progress in cGAN training has been made through regularization. For example, Ohayon et al. [29] proposed to enforce correctness in the generated $\boldsymbol{y}$-conditional mean using a novel form of L2 regularization. Later, Bendel et al. [19] proposed to enforce correctness in the generated $\boldsymbol{y}$-conditional mean *and* trace-covariance using L1 regularization plus a correctly weighted standard-deviation (SD) reward, and gave evidence that their "rcGAN" competes with contemporary diffusion samplers on MRI and inpainting tasks. Soon after, Man et al. [30] showed that L2 regularization and a variance reward are effective when training cGANs for JPEG decoding. More details are given in Section 2.

In a separate line of work, Nehme et al. [31] trained a Neural Posterior Principal Components (NPPC) network to directly estimate the eigenvectors and eigenvalues of the $\boldsymbol{y}$-conditional covariance matrix, which allows powerful insights into the nature of uncertainty in an inverse problem.

**Our contributions.** Inspired by regularized cGANs and NPPC, we propose a novel "pcaGAN" that encourages correctness in the $K$ principal components of the $\boldsymbol{y}$-conditional covariance matrix, as well as the $\boldsymbol{y}$-conditional mean and trace covariance, when sampling from the posterior. We demonstrate that our pcaGAN outperforms existing cGANs [16, 17, 19, 29] and diffusion models [22, 25–27] on posterior sampling tasks like denoising, large-scale inpainting, and accelerated MRI recovery, while sampling orders-of-magnitude faster than those diffusion models. We also demonstrate that pcaGAN recovers the principal components more accurately than NPPC using approximately the same runtime.

## 2  Background

Our approach builds on the rcGAN regularization framework from [19], which itself builds on the cGAN framework from [16], both of which we now summarize. Let $\mathcal{X}$, $\mathcal{Y}$, and $\mathcal{Z}$ denote the domains of $\boldsymbol{x}$, $\boldsymbol{z}$, and $\boldsymbol{y}$, respectively. The goal is to design a generator $\mathcal{G}_{\boldsymbol{\theta}} : \mathcal{Z} \times \mathcal{Y} \to \mathcal{X}$ where, for a given $\boldsymbol{y}$, the random $\widehat{\boldsymbol{x}} = G_{\boldsymbol{\theta}}(\boldsymbol{z}, \boldsymbol{y})$ induced by the code vector $\boldsymbol{z} \sim p_{\mathsf{z}}$ (with $\boldsymbol{z}$ independent of $\boldsymbol{y}$) has a distribution that is close to the true posterior $p_{\mathsf{x}|\mathsf{y}}(\cdot, \boldsymbol{y})$ in the Wasserstein-1 distance, given by

$$W_1\big(p_{\mathsf{x}|\mathsf{y}}(\cdot, \boldsymbol{y}), p_{\widehat{\mathsf{x}}|\mathsf{y}}(\cdot, \boldsymbol{y})\big) = \sup_{D \in L_1} \mathrm{E}_{\mathsf{x}|\mathsf{y}}\{D(\boldsymbol{x}, \boldsymbol{y})\} - \mathrm{E}_{\widehat{\mathsf{x}}|\mathsf{y}}\{D(\widehat{\boldsymbol{x}}, \boldsymbol{y})\}. \tag{1}$$

Here, $D : \mathcal{X} \times \mathcal{Y} \to \mathbb{R}$ is a "critic" or "discriminator" that tries to distinguish between the true $\boldsymbol{x}$ and generated $\widehat{\boldsymbol{x}}$ given $\boldsymbol{y}$, and $L_1$ denotes the set of 1-Lipschitz functions. A loss is constructed by averaging (1) over $\boldsymbol{y} \sim p_{\mathsf{y}}$, which takes the form

$$\mathrm{E}_{\mathsf{y}}\big\{W_1\big(p_{\mathsf{x}|\mathsf{y}}(\cdot, \boldsymbol{y}), p_{\widehat{\mathsf{x}}|\mathsf{y}}(\cdot, \boldsymbol{y})\big)\big\} = \sup_{D \in L_1} \mathrm{E}_{\mathsf{x},\mathsf{z},\mathsf{y}}\{D(\boldsymbol{x}, \boldsymbol{y}) - D(G_{\boldsymbol{\theta}}(\boldsymbol{z}, \boldsymbol{y}), \boldsymbol{y})\}, \tag{2}$$

using the fact that the expectation commutes with the supremum [16]. $D$ is then implemented as a neural network $D_{\boldsymbol{\phi}}$ with parameters $\boldsymbol{\phi}$. Finally, $(\boldsymbol{\theta}, \boldsymbol{\phi})$ are learned by minimizing

$$\mathcal{L}_{\mathsf{adv}}(\boldsymbol{\theta}, \boldsymbol{\phi}) \triangleq \mathrm{E}_{\mathsf{x},\mathsf{z},\mathsf{y}}\{D_{\boldsymbol{\phi}}(\boldsymbol{x}, \boldsymbol{y}) - D_{\boldsymbol{\phi}}(G_{\boldsymbol{\theta}}(\boldsymbol{z}, \boldsymbol{y}), \boldsymbol{y})\} \tag{3}$$

over $\boldsymbol{\theta}$ and minimizing $-\mathcal{L}_{\mathsf{adv}}(\boldsymbol{\theta}, \boldsymbol{\phi}) + \mathcal{L}_{\mathsf{gp}}(\boldsymbol{\phi})$ over $\boldsymbol{\phi}$, where $\mathcal{L}_{\mathsf{gp}}(\boldsymbol{\phi})$ is a gradient penalty that encourages $D_{\boldsymbol{\phi}} \in L_1$ [32]. In practice, the expectation in (3) is approximated by a sample average over the training data $\{(\boldsymbol{x}_t, \boldsymbol{y}_t)\}$.

In the typical case that the training data includes only a single image $\boldsymbol{x}_t$ for each measurement vector $\boldsymbol{y}_t$, minimizing $\mathcal{L}_{\mathsf{adv}}(\boldsymbol{\theta}, \boldsymbol{\phi})$ alone does not encourage the generator to produce diverse samples. Rather, it leads to a form of mode collapse where the code $\boldsymbol{z}$ is ignored. In [16], Adler and Öktem proposed a two-sample discriminator that encourages diverse generator outputs without compromising the Wasserstein objective (2). In [19], Bendel et al. instead proposed to regularize $\mathcal{L}_{\mathsf{adv}}(\boldsymbol{\theta}, \boldsymbol{\phi})$ in a way that encourages correct posterior means and trace-covariances, i.e.,

$$\boldsymbol{\mu}_{\widehat{\mathsf{x}}|\mathsf{y}} = \boldsymbol{\mu}_{\mathsf{x}|\mathsf{y}} \qquad \text{for } \boldsymbol{\mu}_{\widehat{\mathsf{x}}|\mathsf{y}} \triangleq \mathrm{E}\{\widehat{\boldsymbol{x}}|\boldsymbol{y}\} \qquad \text{and } \boldsymbol{\mu}_{\mathsf{x}|\mathsf{y}} \triangleq \mathrm{E}\{\boldsymbol{x}|\boldsymbol{y}\} \tag{4}$$

$$\mathrm{tr}(\boldsymbol{\Sigma}_{\widehat{\mathsf{x}}|\mathsf{y}}) = \mathrm{tr}(\boldsymbol{\Sigma}_{\mathsf{x}|\mathsf{y}}) \qquad \text{for } \boldsymbol{\Sigma}_{\widehat{\mathsf{x}}|\mathsf{y}} \triangleq \mathrm{Cov}\{\widehat{\boldsymbol{x}}|\boldsymbol{y}\} \qquad \text{and } \boldsymbol{\Sigma}_{\mathsf{x}|\mathsf{y}} \triangleq \mathrm{Cov}\{\boldsymbol{x}|\boldsymbol{y}\}. \tag{5}$$

To do this, [19] replaced $\mathcal{L}_{\mathsf{adv}}(\boldsymbol{\theta}, \boldsymbol{\phi})$ with the regularized adversarial loss

$$\mathcal{L}_{\mathsf{rcGAN}}(\boldsymbol{\theta}, \boldsymbol{\phi}) \triangleq \beta_{\mathsf{adv}} \mathcal{L}_{\mathsf{adv}}(\boldsymbol{\theta}, \boldsymbol{\phi}) + \mathcal{L}_{1,P_{\mathsf{rc}}}(\boldsymbol{\theta}) - \beta_{\mathsf{SD}} \mathcal{L}_{\mathsf{SD},P_{\mathsf{rc}}}(\boldsymbol{\theta}), \tag{6}$$

which involves the $P_{\mathsf{rc}}$-sample supervised-$\ell_1$ loss and standard-deviation (SD) reward terms

$$\mathcal{L}_{1,P}(\boldsymbol{\theta}) \triangleq \mathrm{E}_{\mathsf{x},\mathsf{z}_1,\dots,\mathsf{z}_P,\mathsf{y}} \left\{ \| \boldsymbol{x} - \widehat{\boldsymbol{x}}_{(P)} \|_1 \right\} \tag{7}$$

$$\mathcal{L}_{\mathsf{SD},P}(\boldsymbol{\theta}) \triangleq \sum_{i=1}^{P} \mathrm{E}_{\mathsf{z}_1,\dots,\mathsf{z}_P,\mathsf{y}} \left\{ \| \widehat{\boldsymbol{x}}_i - \widehat{\boldsymbol{x}}_{(P)} \|_1 \right\}, \tag{8}$$

where typically $P_{\mathsf{rc}} = 2$. Here, $\{\widehat{\boldsymbol{x}}_i\}$ are the generated samples and $\widehat{\boldsymbol{x}}_{(P)}$ is their $P$-sample average:

$$\widehat{\boldsymbol{x}}_i \triangleq G_{\boldsymbol{\theta}}(\boldsymbol{z}_i, \boldsymbol{y}) \text{ for } i = 1, \dots, P_{\mathsf{rc}} \quad \text{and} \quad \widehat{\boldsymbol{x}}_{(P)} \triangleq \frac{1}{P} \sum_{i=1}^{P} \widehat{\boldsymbol{x}}_i. \tag{9}$$

The reward weight $\beta_{\mathsf{SD}}$ in (6) is automatically adjusted to accomplish (5) during training [19]. We note that the $\mathcal{L}_{1,P}(\boldsymbol{\theta})$ regularization proposed in [19] is closely related to the regularization $\mathcal{L}_{2,P}(\boldsymbol{\theta}) \triangleq \mathrm{E}_{\mathsf{x},\mathsf{z}_1,\dots,\mathsf{z}_P,\mathsf{y}}\{\|\boldsymbol{x} - \widehat{\boldsymbol{x}}_{(P)}\|_2^2\}$ proposed earlier by Ohayon et al. in [29]. A detailed discussion of the advantages and disadvantages of various cGAN regularizations can be found in [19].

## 3 Proposed method

Whereas rcGAN aimed for correctness in the posterior mean and posterior trace-covariance statistics, our proposed pcaGAN *also* aims for correctness in the $K$ principal components of the posterior covariance matrix $\boldsymbol{\Sigma}_{\widehat{\mathsf{x}}|\mathsf{y}}$, where $K$ is user-selectable. To do this, pcaGAN adds two additional regularization terms to the rcGAN objective:

$$\mathcal{L}_{\mathsf{pcaGAN}}(\boldsymbol{\theta}, \boldsymbol{\phi}) \triangleq \mathcal{L}_{\mathsf{rcGAN}}(\boldsymbol{\theta}, \boldsymbol{\phi}) + \beta_{\mathsf{pca}} \mathcal{L}_{\mathsf{evec}}(\boldsymbol{\theta}) + \beta_{\mathsf{pca}} \mathcal{L}_{\mathsf{eval}}(\boldsymbol{\theta}) \tag{10}$$

$$\mathcal{L}_{\mathsf{evec}}(\boldsymbol{\theta}) \triangleq -\mathrm{E}_{\mathsf{y}} \left\{ \mathrm{E}_{\mathsf{x},\mathsf{z}_1,\dots,\mathsf{z}_P|\mathsf{y}} \left\{ \sum_{k=1}^{K} [\widehat{\boldsymbol{v}}_k^\top (\boldsymbol{x} - \boldsymbol{\mu}_{\mathsf{x}|\mathsf{y}})]^2 \big| \boldsymbol{y} \right\} \right\} \tag{11}$$

$$\mathcal{L}_{\mathsf{eval}}(\boldsymbol{\theta}) \triangleq \mathrm{E}_{\mathsf{y}} \left\{ \mathrm{E}_{\mathsf{x},\mathsf{z}_1,\dots,\mathsf{z}_P|\mathsf{y}} \left\{ \sum_{k=1}^{K} \left( 1 - \lambda_k / \widehat{\lambda}_k \right)^2 \big| \boldsymbol{y} \right\} \right\}. \tag{12}$$

Here, $\{(\widehat{\boldsymbol{v}}_k, \widehat{\lambda}_k)\}_{k=1}^{K}$ denote the principal eigenvectors and eigenvalues of the $\boldsymbol{\theta}$-dependent generated covariance matrix $\boldsymbol{\Sigma}_{\widehat{\mathsf{x}}|\mathsf{y}}$ and $\{(\boldsymbol{v}_k, \lambda_k)\}_{k=1}^{K}$ denote the principal eigenvectors and eigenvalues of the true covariance matrix $\boldsymbol{\Sigma}_{\mathsf{x}|\mathsf{y}}$. Because (11) is the classical PCA objective [33], minimizing $\mathcal{L}_{\mathsf{evec}}(\boldsymbol{\theta})$ over $\boldsymbol{\theta}$ will drive the generated principal eigenvector $\widehat{\boldsymbol{v}}_k$ towards the true principal eigenvector $\boldsymbol{v}_k$ for each $k = 1, \dots, K$. Likewise, minimizing $\mathcal{L}_{\mathsf{eval}}(\boldsymbol{\theta})$ over $\boldsymbol{\theta}$ will drive the generated principal eigenvalue $\widehat{\lambda}_k$ towards the true principal eigenvalue $\lambda_k$ for each $k = 1, \dots, K$. Based on our experiments, putting $\widehat{\lambda}_k$ in the denominator works better than the numerator and the squared error in (12) works better than an absolute value.

In practice, the expectations in (11)-(12) are replaced by sample averages over the training data. In the typical case that the training data includes only a single image $\boldsymbol{x}_t$ for each measurement vector $\boldsymbol{y}_t$, the quantities $\boldsymbol{\mu}_{\mathsf{x}|\mathsf{y}}$ and $\{\lambda_k\}$ in (11)-(12) are unknown and non-trivial to estimate for each $\boldsymbol{y}_t$. Hence, when training the pcaGAN, we approximate them with learned quantities. This must be done carefully, however. For example, if $\boldsymbol{\mu}_{\mathsf{x}|\mathsf{y}}$ in (11) was simply replaced by the $\boldsymbol{\theta}$-dependent quantity $\boldsymbol{\mu}_{\widehat{\mathsf{x}}|\mathsf{y}}$, then minimizing $\mathcal{L}_{\mathsf{evec}}(\boldsymbol{\theta})$ over $\boldsymbol{\theta}$ would encourage $\boldsymbol{\mu}_{\widehat{\mathsf{x}}|\mathsf{y}}$ to become overly large in order to drive $\mathcal{L}_{\mathsf{evec}}(\boldsymbol{\theta})$ to a large negative value.

Algorithm 1 details our proposed approach to training the pcaGAN. In particular, it describes the steps used to perform a single update of the generator parameters $\boldsymbol{\theta}$ based on the training batch $\{(\boldsymbol{x}_b, \boldsymbol{y}_b)\}_{b=1}^{B}$. Before diving into the details, we offer a brief summary of Algorithm 1. For the initial epochs, the rcGAN objective $\mathcal{L}_{\mathsf{rcGAN}}$ alone is optimized, which allows the generated posterior mean $\boldsymbol{\mu}_{\widehat{\mathsf{x}}|\mathsf{y}}$ to converge to the vicinity of $\boldsymbol{\mu}_{\mathsf{x}|\mathsf{y}}$. Starting at $E_{\mathsf{evec}}$ epochs, the $\mathcal{L}_{\mathsf{evec}}(\boldsymbol{\theta})$ regularization from (11) is added, but with $\boldsymbol{\mu}_{\mathsf{x}|\mathsf{y}}$ approximated as $\mathtt{StopGrad}(\boldsymbol{\mu}_{\widehat{\mathsf{x}}|\mathsf{y}})$. The use of $\mathtt{StopGrad}$ forces $\mathcal{L}_{\mathsf{evec}}(\boldsymbol{\theta})$ to be minimized by manipulating the eigenvectors $\{\widehat{\boldsymbol{v}}_k\}_{k=1}^{K}$ and not the generated posterior mean $\boldsymbol{\mu}_{\widehat{\mathsf{x}}|\mathsf{y}}$. These eigenvectors are computed using an SVD of centered approximate-posterior samples. To reduce the computational burden imposed by this SVD, a "lazy regularization" [34] approach is adopted, which computes $\mathcal{L}_{\mathsf{evec}}(\boldsymbol{\theta})$ only once every $M$ training steps. Training proceeds in this manner until the eigenvectors $\{\widehat{\boldsymbol{v}}_k\}$ converge. Starting at $E_{\mathsf{eval}}$ epochs, the $\mathcal{L}_{\mathsf{eval}}(\boldsymbol{\theta})$ regularization from (12) is added, but with $\lambda_k$ approximated as

$$\lambda_k \approx \mathtt{StopGrad}\big(\tfrac{1}{1+P_{\mathsf{pca}}} \big\| \widehat{\boldsymbol{v}}_k^\top [\boldsymbol{x}_b - \boldsymbol{\mu}_{\widehat{\mathsf{x}}|\mathsf{y}}, \widehat{\boldsymbol{x}}_1 - \boldsymbol{\mu}_{\widehat{\mathsf{x}}|\mathsf{y}}, \dots, \widehat{\boldsymbol{x}}_{P_{\mathsf{pca}}} - \boldsymbol{\mu}_{\widehat{\mathsf{x}}|\mathsf{y}}] \big\|^2 \big), \tag{13}$$

---

**Algorithm 1** pcaGAN generator-training iteration

---

**Require:** number of estimated eigen-components $K$, number of samples used for eigenvector and eigenvalue regularization $P_{\mathsf{pca}}$, number of samples used for rcGAN regularization $P_{\mathsf{rc}}$, epoch at which to involve eigenvector regularization $E_{\mathsf{evec}}$, epoch at which to involve eigenvalue regularization $E_{\mathsf{eval}}$, lazy update period $M$, adversarial loss weight $\beta_{\mathsf{adv}}$, std regularization weight $\beta_{\mathsf{SD}}$, eigenvector and eigenvalue regularization weight $\beta_{\mathsf{pca}}$, training batch $\{(\boldsymbol{x}_b, \boldsymbol{y}_b)\}_{b=1}^{B}$, current model parameters $\boldsymbol{\theta}$, current training epoch $e$, the current training step $s$

1:   $\mathcal{L}(\boldsymbol{\theta}) \leftarrow 0$
2:
3:   **for** $b = 1, \ldots, B$ **do**
4:       $\boldsymbol{z}_i \sim \mathcal{N}(\boldsymbol{0}, \boldsymbol{I})$ for $i = 1, \ldots, P_{\mathsf{rc}}$
5:       $\widehat{\boldsymbol{x}}_i \leftarrow G_{\boldsymbol{\theta}}(\boldsymbol{z}_i, \boldsymbol{y}_b)$ for $i = 1, \ldots, P_{\mathsf{rc}}$
6:       $\mathcal{L}(\boldsymbol{\theta}) \leftarrow \mathcal{L}(\boldsymbol{\theta}) - \beta_{\mathsf{adv}} \sum_{i=1}^{P_{\mathsf{rc}}} D_{\boldsymbol{\phi}}(\widehat{\boldsymbol{x}}_i, \boldsymbol{y}_b)$
7:       $\widehat{\boldsymbol{x}}_{(P_{\mathsf{rc}})} = \frac{1}{P_{\mathsf{rc}}} \sum_{i=1}^{P_{\mathsf{rc}}} \widehat{\boldsymbol{x}}_i$
8:       $\mathcal{L}(\boldsymbol{\theta}) \leftarrow \mathcal{L}(\boldsymbol{\theta}) + \|\boldsymbol{x}_b - \widehat{\boldsymbol{x}}_{(P_{\mathsf{rc}})}\|_1 - \beta_{\mathsf{SD}} \sum_{i=1}^{P_{\mathsf{rc}}} \mathrm{E}_{\boldsymbol{z}_1, \ldots, \boldsymbol{z}_\mathsf{P}, \mathsf{y}} \left\{ \|\widehat{\boldsymbol{x}}_i - \widehat{\boldsymbol{x}}_{(P_{\mathsf{rc}})}\|_1 \right\}$
9:
10:      **if** $e \geq E_{\mathsf{evec}}$ and $s \bmod M = 0$ **then**
11:         $\boldsymbol{z}_j \sim \mathcal{N}(\boldsymbol{0}, \boldsymbol{I})$ for $j = 1, \ldots, P_{\mathsf{pca}}$
12:         $\widehat{\boldsymbol{x}}_j \leftarrow G_{\boldsymbol{\theta}}(\boldsymbol{z}_j, \boldsymbol{y}_b)$ for $j = 1, \ldots, P_{\mathsf{pca}}$
13:         $\widehat{\boldsymbol{\mu}} \leftarrow \mathtt{StopGrad}(\frac{1}{P_{\mathsf{pca}}} \sum_{j=1}^{P_{\mathsf{pca}}} \widehat{\boldsymbol{x}}_j)$
14:         $\widehat{\boldsymbol{U}} \widehat{\boldsymbol{S}} \widehat{\boldsymbol{V}}^{\top} \leftarrow \mathrm{SVD}([\widehat{\boldsymbol{x}}_1 - \widehat{\boldsymbol{\mu}}, \ldots, \widehat{\boldsymbol{x}}_{P_{\mathsf{pca}}} - \widehat{\boldsymbol{\mu}}]^{\top})$
15:         $\widehat{\boldsymbol{v}}_k \leftarrow [\widehat{\boldsymbol{V}}]_{:,k}$ for $k = 1, \ldots, K$
16:         $\mathcal{L}(\boldsymbol{\theta}) \leftarrow \mathcal{L}(\boldsymbol{\theta}) - \beta_{\mathsf{pca}} \sum_{k=1}^{K} [\widehat{\boldsymbol{v}}_k^{\top} (\boldsymbol{x}_b - \widehat{\boldsymbol{\mu}})]^2$
17:      **end if**
18:
19:      **if** $e \geq E_{\mathsf{eval}}$ and $s \bmod M = 0$ **then**
20:         $\widehat{\lambda}_k \leftarrow [\widehat{\boldsymbol{S}}]_{kk}^2$ for $k = 1, \ldots, K$
21:         $\widetilde{\boldsymbol{X}} \leftarrow [\boldsymbol{x}_b - \widehat{\boldsymbol{\mu}}, \widehat{\boldsymbol{x}}_1 - \widehat{\boldsymbol{\mu}}, \widehat{\boldsymbol{x}}_2 - \widehat{\boldsymbol{\mu}}, \ldots, \widehat{\boldsymbol{x}}_{P_{\mathsf{pca}}} - \widehat{\boldsymbol{\mu}}]^{\top}$
22:         $\mathcal{L}(\boldsymbol{\theta}) \leftarrow \mathcal{L}(\boldsymbol{\theta}) + \beta_{\mathsf{pca}} \sum_{k=1}^{K} \left(1 - \frac{1}{\widehat{\lambda}_k} \mathtt{StopGrad}(\frac{1}{P_{\mathsf{pca}}+1} \|\widehat{\boldsymbol{v}}_k^{\top} \widetilde{\boldsymbol{X}}\|^2)\right)^2$
23:      **end if**
24:   **end for**
25:
26:   $\boldsymbol{\theta} \leftarrow \mathrm{Adam}(\boldsymbol{\theta}, \nabla \mathcal{L}(\boldsymbol{\theta}))$

---

where $\mathtt{StopGrad}$ is used so that the optimization focuses on $\{\widehat{\lambda}_k\}$. The rational behind (13) is that, when $\widehat{\boldsymbol{v}}_k = \boldsymbol{v}_k$ and $\boldsymbol{\mu}_{\widehat{\mathsf{x}}|\mathsf{y}} = \boldsymbol{\mu}_{\mathsf{x}|\mathsf{y}}$, the terms $[\widehat{\boldsymbol{v}}_k^{\top} (\boldsymbol{x}_b - \boldsymbol{\mu}_{\widehat{\mathsf{x}}|\mathsf{y}})]^2$ and $[\widehat{\boldsymbol{v}}_k^{\top} (\widehat{\boldsymbol{x}}_j - \boldsymbol{\mu}_{\widehat{\mathsf{x}}|\mathsf{y}})]^2 \; \forall j$ all equal $\lambda_k$ in $\boldsymbol{y}_b$-conditional expectation. This expectation is approximated using a $(1 + P_{\mathsf{pca}})$-term sample average in (13) via the squared norm. The eigenvalues $\{\widehat{\lambda}_k\}$ in (12) are computed using the previously described SVD and the regularization schedule is again $M$-lazy.

We now provide additional details on Algorithm 1. After the loss is initialized in line 1, the following steps are executed for each measurement vector $\boldsymbol{y}_b$ in the batch. First, approximate posterior samples $\{\widehat{\boldsymbol{x}}_i\}_{i=1}^{P_{\mathsf{rc}}}$ are generated in line 5, where $P_{\mathsf{rc}} = 2$ as per the suggestion in [19]. Using these samples, the adversarial component of the loss is added in line 6 and the rcGAN regularization is added in line 8. Starting at epoch $E_{\mathsf{evec}}$, lines 11-16 are executed whenever the training iteration is a multiple of $M$. Nominally, $E_{\mathsf{evec}}$ is set where the validation PSNR of $\widehat{\boldsymbol{\mu}}$ (an empirical approximation of $\boldsymbol{\mu}_{\widehat{\mathsf{x}}|\mathsf{y}}$) stabilizes and $M = 100$. Within those lines, samples $\{\widehat{\boldsymbol{x}}_j\}_{j=1}^{P_{\mathsf{pca}}}$ are generated in line 12 (where nominally $P_{\mathsf{pca}} = 10K$), their sample mean is computed in line 13, and the SVD of the centered samples is computed in line 14. The top $K$ right singular vectors are then extracted in line 15 in order to construct the $\mathcal{L}_{\mathsf{evec}}(\boldsymbol{\theta})$ regularization, which is added to the overall generator loss $\mathcal{L}(\boldsymbol{\theta})$ in line 16. Starting at epoch $E_{\mathsf{eval}}$, where nominally $E_{\mathsf{eval}} = E_{\mathsf{evec}} + 25$, lines 20-22 are executed whenever the training iteration is a multiple of $M$. In line 20, the top $K$ eigenvalues $\{\widehat{\lambda}_k\}$ are constructed from the previously computed singular values and, in line 22, the $\mathcal{L}_{\mathsf{eval}}(\boldsymbol{\theta})$ regularization is constructed

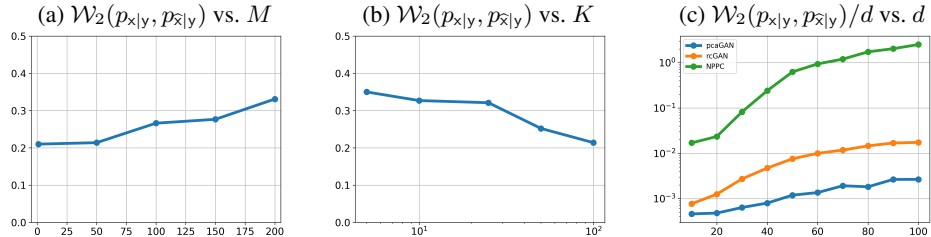

Figure 1: Gaussian experiment. Wasserstein-2 distance versus (a) lazy update period $M$ for pcaGAN with $d = 100 = K$, (b) estimated eigen-components $K$ for pcaGAN with $d = 100$ and $M = 100$, and (c) problem dimension $d$ for all methods under test with $K = d$ and $M = 100$.

and added to the overall training loss. The construction of $\mathcal{L}_{\text{eval}}(\boldsymbol{\theta})$ was previously described around (13). Finally, once the losses for all batch elements have been incorporated, the gradient $\nabla \mathcal{L}(\boldsymbol{\theta})$ is computed using back-propagation and a gradient-descent step of $\boldsymbol{\theta}$ is performed using the Adam optimizer [35] in line 26.

## 4 Numerical experiments

We now present experiments with Gaussian data, MNIST denoising, MRI, and FFHQ face inpainting. Additional implementation and training details for each experiment are provided in Appendix D.

### 4.1 Recovering synthetic Gaussian data

Here our goal is to recover $\boldsymbol{x} \sim \mathcal{N}(\boldsymbol{\mu}_{\text{x}}, \boldsymbol{\Sigma}_{\text{x}}) \in \mathbb{R}^d$ from $\boldsymbol{y} = \boldsymbol{M}\boldsymbol{x} + \boldsymbol{w} \in \mathbb{R}^d$, where $\boldsymbol{M}$ masks $\boldsymbol{x}$ at even indices and noise $\boldsymbol{w} \sim \mathcal{N}(\boldsymbol{0}, \sigma^2\boldsymbol{I})$ is independent of $\boldsymbol{x}$ with $\sigma^2 = 0.001$. Since $\boldsymbol{x}$ and $\boldsymbol{y}$ are jointly Gaussian, the posterior is Gaussian posterior with $\boldsymbol{\mu}_{\text{x|y}} = \boldsymbol{\mu}_{\text{x}} + \boldsymbol{\Sigma}_{\text{xy}}\boldsymbol{\Sigma}_{\text{y}}^{-1}(\boldsymbol{y} - \boldsymbol{\mu}_{\text{y}})$ and $\boldsymbol{\Sigma}_{\text{x|y}} = \boldsymbol{\Sigma}_{\text{x}} - \boldsymbol{\Sigma}_{\text{xy}}\boldsymbol{\Sigma}_{\text{y}}^{-1}\boldsymbol{\Sigma}_{\text{yx}}$, where $\boldsymbol{\mu}_{\text{x}}, \boldsymbol{\mu}_{\text{y}}, \boldsymbol{\Sigma}_{\text{x}}, \boldsymbol{\Sigma}_{\text{y}}$ are marginal and $\boldsymbol{\Sigma}_{\text{xy}}, \boldsymbol{\Sigma}_{\text{yx}}$ are joint statistics.

We generate random $\boldsymbol{\mu}_{\text{x}} \sim \mathcal{N}(\boldsymbol{0}, \boldsymbol{I})$ and $\boldsymbol{\Sigma}_{\text{x}}$ with half-normal eigenvalues $\lambda_k$ (see additional details in App. A), and consider a sequence of problem sizes $d = 10, 20, 30, \ldots, 100$. For each $d$, we generate 70 000 training, 20 000 validation, and 10 000 test samples. The generator and discriminator are simple multilayer perceptrons (see App. D.1) trained for 100 epochs with $K = d$, $E_{\text{evec}} = 10$, $\beta_{\text{adv}} = 10^{-5}$, and $\beta_{\text{pca}} = 10^{-2}$.

**Competitors.** We compare the proposed pcaGAN to rcGAN [19] and NPPC [31]. rcGAN uses the same generator and discriminator architectures as pcaGAN and is trained according to (6) with $\beta_{\text{adv}} = 10^{-5}$ and $P_{\text{rc}} = 2$. For NPPC, we use the authors' implementation [36] with $K = d$ and some minor modifications to work with vector data. To evaluate performance, we use the Wasserstein-2 (W2) distance between $p_{\text{x|y}}$ and $\widehat{p_{\text{x|y}}}$, which in the Gaussian case reduces to

$$\mathcal{W}_2(p_{\text{x|y}}, \widehat{p_{\text{x|y}}}) = \|\boldsymbol{\mu}_{\text{x|y}} - \widehat{\boldsymbol{\mu}_{\text{x|y}}}\|_2^2 + \text{tr}\left[\boldsymbol{\Sigma}_{\text{x|y}} + \widehat{\boldsymbol{\Sigma}_{\text{x|y}}} - 2(\boldsymbol{\Sigma}_{\text{x|y}}^{1/2}\widehat{\boldsymbol{\Sigma}_{\text{x|y}}}\boldsymbol{\Sigma}_{\text{x|y}}^{1/2})^{1/2}\right]. \quad (14)$$

For the cGANs, we compute $\widehat{\boldsymbol{\mu}_{\text{x|y}}}$ and $\widehat{\boldsymbol{\Sigma}_{\text{x|y}}}$ empirically from $10d$ samples, while for NPPC we use the conditional mean, eigenvalues, and eigenvectors returned by the approach.

**Results.** Figure 1a examines the impact of the lazy update period $M$ on pcaGAN's W2 distance at $d = 100$ with $K = d$. Based on this figure, to balance performance with training overhead, we set $M = 100$ for all future experiments. Figure 1b examines the impact of $K$ on W2 distance for the pcaGAN with $d = 100$. It shows that using $K < d$ causes a relatively mild increase in W2 distance, as expected due to the half-normal distribution on the true eigenvalues $\lambda_k$. Figure 1c shows that the proposed pcaGAN outperforms rcGAN and NPPC in W2 distance for all problem sizes $d$.

### 4.2 MNIST denoising

Now our goal is to recover an MNIST digit $\boldsymbol{x} \in [0, 1]^{28 \times 28}$ from noisy measurements $\boldsymbol{y} = \boldsymbol{x} + \boldsymbol{w}$ with $\boldsymbol{w} \sim \mathcal{N}(\boldsymbol{0}, \boldsymbol{I})$. We randomly split the MNIST training fold into 50 000 training and 10 000

Table 1: Average MNIST denoising results.

| Model | rMSE↓ | REM$_5$↓ | CFID↓ | Time(128)↓ |
|---|---|---|---|---|
| NPPC (Nehme et al. [31]) | **3.94** | 3.63 | – | **112 ms** |
| rcGAN (Bendel et al. [19]) | 4.04 | 3.41 | 63.44 | 118 ms |
| pcaGAN (Ours, $K = 5$) | 4.02 | 3.31 | 61.48 | 118 ms |
| pcaGAN (Ours, $K = 10$) | 4.02 | **3.25** | **60.16** | 118 ms |

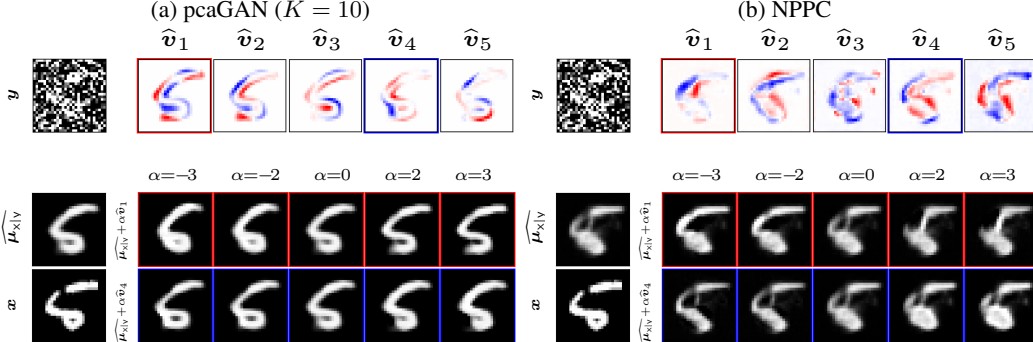

Figure 2: For (a) pcaGAN and (b) NPPC, this figure shows the true image $x$, noisy measurements $y$, the conditional mean $\widehat{\boldsymbol{\mu}_{x|y}}$, principal eigenvectors $\{\widehat{\boldsymbol{v}}_k\}$, and two perturbations of $\widehat{\boldsymbol{\mu}_{x|y}}$.

validation images, and we use the entire MNIST fold set for testing. For pcaGAN and rcGAN, we use a UNet [37] generator and the encoder portion of the same UNet followed by one dense layer as the discriminator. pcaGAN was trained for 125 epochs with $E_{\text{evec}} = 25$, $\beta_{\text{adv}} = 10^{-5}$, $\beta_{\text{pca}} = 10^{-1}$, and $K \in \{5, 10\}$.

**Competitors.** We again compare the proposed pcaGAN to rcGAN and NPPC. For rcGAN we used the same generator and discriminator architectures as pcaGAN and trained according to (6) with $\beta_{\text{adv}} = 10^{-5}$ and $P_{\text{rc}} = 2$. For NPPC, we used the authors' MNIST implementation from [36].

Following the NPPC paper [31], we evaluate performance using root MSE (rMSE) $\text{E}_{x,y}\{\|\boldsymbol{x} - \widehat{\boldsymbol{\mu}_{x|y}}\|_2\}$ and residual error magnitude (REM$_5$) $\text{E}_{x,y}\left\{\left\|(\boldsymbol{I} - \widehat{\boldsymbol{V}}_5\widehat{\boldsymbol{V}}_5^\top)\boldsymbol{e}\right\|_2\right\}$, where $\boldsymbol{e} = \boldsymbol{x} - \widehat{\boldsymbol{\mu}_{x|y}}$ and $\widehat{\boldsymbol{V}}_5$ is an $28^2 \times 5$ matrix whose $k$th column equals the $k$th principal eigenvector $\widehat{\boldsymbol{v}}_k$. For the cGANs, we use $\widehat{\boldsymbol{\mu}_{x|y}} = \widehat{\boldsymbol{x}}_{(P)}$ and compute $\{\widehat{\boldsymbol{v}}_k\}$ from the SVD of a matrix of centered samples $\{\widehat{\boldsymbol{x}}_i\}_{i=1}^P$, both with $P = 100$. For NPPC, we use the conditional means and eigenvectors returned by the approach. For performance evaluation, we also consider Conditional Frechet Inception Distance (CFID) [38] with InceptionV3 features. CFID is analogous to Frechet Inception Distance (FID) [39] but applies to conditional distributions (see App. B for more details).

**Results.** Table 1 shows rMSE, REM$_5$, CFID, and the reconstruction time for a batch of 128 images on the test fold. (NPPC does not generate image samples and so CFID does not apply.) The table shows that the proposed pcaGAN wins in all metrics, except for rMSE where NPPC wins. This is not surprising because NPPC computes $\widehat{\boldsymbol{\mu}_{x|y}}$ using a dedicated network trained to minimize MSE loss. NPPC also generates its eigenvectors slightly quicker than pcaGAN generates samples. Table 1 also shows that pcaGAN performance improves as $K$ increases from 5 to 10, despite the fact that REM$_5$ uses only the top 5 eigenvectors. Figure E.1 shows examples of the 5 principal eigenvectors and posterior mean learned by pcaGAN and NPPC. The eigenvectors of pcaGAN are more structured and less noisy than those of NPPC. Figure E.1 also shows $\widehat{\boldsymbol{\mu}_{x|y}} + \alpha\boldsymbol{v}_k$ for $\alpha \in [-3, 3]$ and $k \in \{1, 4\}$. Additional figures can be found in App. E.1.

### 4.3 Accelerated MRI

We now consider accelerated MRI, where the goal is to recover a complex-valued multicoil image $x$ from masked frequency-domain (i.e., "k-space") measurements $y$. To build the image data $\{\boldsymbol{x}_t\}$, we follow the approach in the rcGAN paper [19], which uses the first 8 slices of all fastMRI [40] T2

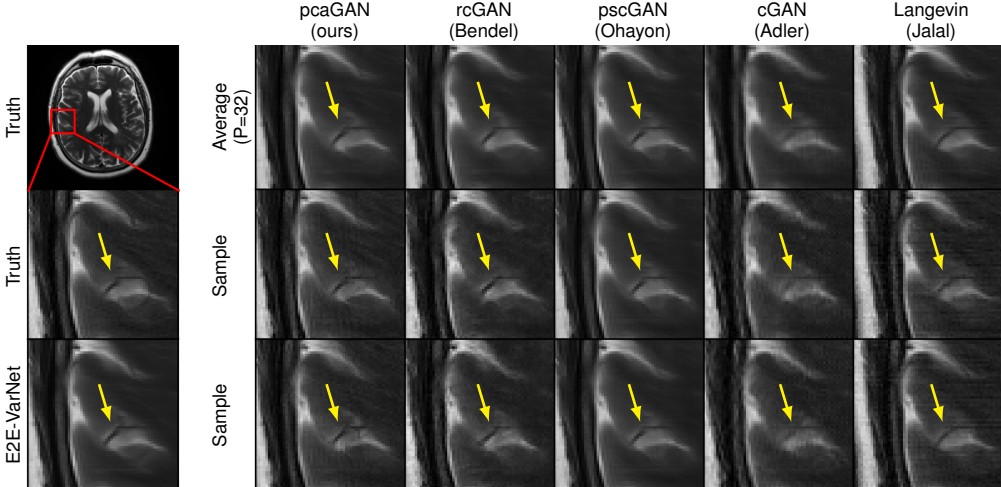

Figure 3: Example MRI recoveries at $R = 8$. Arrows highlight meaningful variations.

Table 2: Average MRI results at acceleration $R \in \{4, 8\}$

| | $R = 4$ | | | | | | $R = 8$ | | | | | |
|---|---|---|---|---|---|---|---|---|---|---|---|---|
| Model | CFID$^1\downarrow$ | CFID$^2\downarrow$ | CFID$^3\downarrow$ | FID$\downarrow$ | APSD | Time (4)$\downarrow$ | CFID$^1\downarrow$ | CFID$^2\downarrow$ | CFID$^3\downarrow$ | FID$\downarrow$ | APSD | Time (4)$\downarrow$ |
| E2E-VarNet (Sriram et al. [43]) | 16.08 | 13.07 | 10.26 | 38.88 | 0.0 | 310ms | 36.86 | 29.90 | 23.82 | 44.04 | 0.0 | 316ms |
| Langevin (Jalal et al. [22]) | 33.05 | - | - | 31.43 | 5.9e-6 | 14 min | 48.59 | - | - | 52.62 | 7.6e-6 | 14 min |
| cGAN (Adler & Öktem [16]) | 19.00 | 12.05 | 7.00 | 29.77 | 3.9e-6 | **217 ms** | 59.94 | 40.24 | 26.10 | 31.81 | 7.7e-6 | **217 ms** |
| pscGAN (Ohayon et al. [29]) | 13.74 | 10.56 | 7.53 | 37.28 | 7.2e-8 | **217 ms** | 39.67 | 31.81 | 24.06 | 43.39 | 7.7e-7 | **217 ms** |
| rcGAN (Bendel et al. [19]) | 9.71 | 5.27 | 1.69 | 25.62 | 3.8e-6 | **217 ms** | 24.04 | 13.20 | 3.83 | 28.43 | 7.6e-6 | **217 ms** |
| pcaGAN (Ours) | **8.78** | **4.48** | **1.29** | **25.02** | 4.4e-6 | **217 ms** | **21.65** | **11.47** | **3.21** | **28.35** | 6.5e-6 | **217 ms** |

brain volumes with at least 8 coils, crops to $384 \times 384$ pixels, and compresses to 8 virtual coils [41]. This yields 12 200 training, 2 376 testing, and 784 validation images. To create each $\boldsymbol{y}_t$, we transform $\boldsymbol{x}_t$ to the k-space, subsample using the Cartesian GRO mask [42] at accelerations $R = 4$ and $R = 8$, and transform the zero-filled k-space measurements back to the image domain.

We train pcaGAN for 100 epochs with $K = 1$, $E_{\text{evec}} = 25$, $\beta_{\text{adv}} = 10^{-5}$, and $\beta_{\text{pca}} = 10^{-2}$ and select the final model using validation CFID computed with VGG-16 features.

**Competitors.** We compare the proposed pcaGAN to rcGAN [19], pscGAN [29], Adler & Öktem's cGAN [16], the Langevin approach [22], and the E2E-VarNet [43]. All cGANs use the generator and discriminator architectures from [19] and enforce data-consistency [44]. For rcGAN and the Langevin approach, we did not modify the authors' implementations from [45] and [46] except to use the GRO sampling mask. For E2E-VarNet, we use the GRO mask, hyperparameters, and training procedure from [22].

Following [19], we convert the multicoil outputs $\widehat{\boldsymbol{x}}_i$ to complex-valued images using SENSE-based coil combining [47] with ESPIRiT-estimated [48] coil sensitivity maps, and compute performance on magnitude images. All feature-based metrics (CFID, FID, LPIPS, DISTS) were computed with AlexNet features to show that pcaGAN does not overfit to the VGG-16 features used for validation. It was shown in [49] that image-quality metrics computed using ImageNet-trained feature generators like AlexNet and VGG-16 perform comparably to metrics computed using MRI-trained feature generators in terms of correlation with radiologists' scores.

**Results.** Table 2 shows CFID, FID, APSD $\triangleq \left(\frac{1}{P}\sum_{i=1}^{P} \frac{1}{N}\|\widehat{\boldsymbol{x}}_{(P)} - \widehat{\boldsymbol{x}}_i\|^2\right)^{1/2}$, and 4-sample generation time for the methods under test. Due to its slow sample-generation time, we evaluate the CFID, FID, and APSD of the Langevin technique [22] using the 72-image test from [19]. But due to the bias of CFID at small sample sizes [38], we evaluate the other methods using all 2 376 test images (CFID$^2$) and again using all 14 576 training and test images (CFID$^3$). Table 2 shows that pcaGAN yields better CFID and FID than the competitors. All cGANs generated samples 3–4 orders-of-magnitude faster than the Langevin approach [22].

Table 3: Average PSNR, SSIM, LPIPS, and DISTS of $\widehat{x}_{(P)}$ versus $P$ for MRI at $R = 8$

| | PSNR↑ | | | | | | SSIM↑ | | | | | |
|---|---|---|---|---|---|---|---|---|---|---|---|---|
| Model | $P$=1 | $P$=2 | $P$=4 | $P$=8 | $P$=16 | $P$=32 | $P$=1 | $P$=2 | $P$=4 | $P$=8 | $P$=16 | $P$=32 |
| E2E-VarNet (Sriram et al. [43]) | **36.49** | - | - | - | - | - | 0.9220 | - | - | - | - | - |
| Langevin (Jalal et al. [22]) | 32.17 | 32.83 | 33.45 | 33.74 | 33.83 | 33.90 | 0.8725 | 0.8919 | 0.9031 | 0.9091 | 0.9120 | 0.9137 |
| cGAN (Adler & Öktem [16]) | 31.31 | 32.31 | 32.92 | 33.26 | 33.42 | 33.51 | 0.8865 | 0.9045 | 0.9103 | 0.9111 | 0.9102 | 0.9095 |
| pscGAN (Ohayon et al. [29]) | 34.89 | 34.90 | 34.90 | 34.90 | 34.91 | 34.92 | 0.9222 | 0.9217 | 0.9213 | 0.9211 | 0.9211 | 0.9210 |
| rcGAN (Bendel et al. [19]) | 32.32 | 33.67 | 34.53 | 35.01 | 35.27 | 35.42 | 0.9030 | 0.9199 | 0.9252 | 0.9257 | 0.9251 | 0.9246 |
| pcaGAN (Ours) | 33.28 | 34.47 | 35.20 | 35.61 | 35.82 | 35.94 | 0.9136 | 0.9257 | **0.9283** | 0.9275 | 0.9262 | 0.9253 |

| | LPIPS↓ | | | | | | DISTS↓ | | | | | |
|---|---|---|---|---|---|---|---|---|---|---|---|---|
| Model | $P$=1 | $P$=2 | $P$=4 | $P$=8 | $P$=16 | $P$=32 | $P$=1 | $P$=2 | $P$=4 | $P$=8 | $P$=16 | $P$=32 |
| E2E-VarNet (Sriram et al. [43]) | 0.0575 | - | - | - | - | - | 0.1253 | - | - | - | - | - |
| Langevin (Jalal et al. [22]) | 0.0769 | 0.0619 | 0.0579 | 0.0589 | 0.0611 | 0.0611 | 0.1341 | 0.1136 | 0.1086 | 0.1119 | 0.1175 | 0.1212 |
| cGAN (Adler & Öktem [16]) | 0.0698 | 0.0614 | 0.0623 | 0.0667 | 0.0704 | 0.0727 | 0.1407 | 0.1262 | 0.1252 | 0.1291 | 0.1334 | 0.1361 |
| pscGAN (Ohayon et al. [29]) | 0.0532 | 0.0536 | 0.0539 | 0.0540 | 0.0534 | 0.0540 | 0.1128 | 0.1143 | 0.1151 | 0.1155 | 0.1157 | 0.1158 |
| rcGAN (Bendel et al. [19]) | 0.0418 | 0.0379 | 0.0421 | 0.0476 | 0.0516 | 0.0539 | 0.0906 | 0.0877 | 0.0965 | 0.1063 | 0.1135 | 0.1177 |
| pcaGAN (Ours) | 0.0358 | **0.0344** | 0.0391 | 0.0442 | 0.0479 | 0.0499 | 0.0804 | **0.0799** | 0.0920 | 0.1026 | 0.1099 | 0.1144 |

Table 4: Average FFHQ inpainting results.

| Model | CFID↓ | FID↓ | LPIPS↓ | Time (40 samples)↓ |
|---|---|---|---|---|
| DPS (Chung et al. [26]) | 7.26 | 2.00 | 0.1245 | 14 min |
| DDNM (Wang et al. [27]) | 11.30 | 3.63 | 0.1409 | 30 s |
| DDRM (Kawar et al. [25]) | 13.17 | 5.36 | 0.1587 | 5 s |
| pscGAN (Ohayon et al. [29]) | 18.44 | 8.40 | 0.1716 | **325 ms** |
| CoModGAN (Zhao et al. [17]) | 7.85 | 2.23 | 0.1290 | **325 ms** |
| rcGAN (Bendel et al. [19]) | 7.51 | 2.12 | 0.1262 | **325 ms** |
| pcaGAN (Ours) | **7.08** | **1.98** | **0.1230** | **325 ms** |

Table 3 shows PSNR, SSIM, LPIPS [50], and DISTS [51] for the $P$-sample average $\widehat{x}_{(P)}$ at $P \in \{1, 2, 4, 8, 16, 32\}$ and $R = 8$. (See App. C for $R = 4$.) It has been shown that DISTS correlates particularly well with radiologist scores [52]. The E2E-VarNet achieves the best PSNR, but the proposed cGAN achieves the best LPIPS and DISTS when $P = 2$ and the best SSIM when $P = 8$. This $P$-dependence is related to the perception-distortion trade-off [53] and consistent with that reported in [19].

Figure 3 shows zoomed versions of two recoveries $\widehat{x}_i$ and the sample average $\widehat{x}_{(P)}$ with $P = 32$. Appendices E.2 and E.3 show additional plots of $\widehat{x}_{(P)}$ that visually demonstrate the perception-distortion trade-off.

## 4.4 Large-scale inpainting

Our final goal is to inpaint a face image with a large randomly generated masked region. For this task, we use $256 \times 256$ FFHQ face images [54] and the mask generation procedure from [17]. We randomly split the FFHQ training fold into 45 000 training and 5 000 validation images, and we use the remaining 20 000 images for testing.

For pcaGAN, we use CoModGAN's [17] generator and discriminator architecture and train for 100 epochs using $K = 2$, $E_{\text{evec}} = 25$, $\beta_{\text{adv}} = 5 \times 10^{-3}$, and $\beta_{\text{pca}} = 10^{-3}$.

**Competitors.** We compare with CoModGAN [17], pscGAN [29], rcGAN [19], and state-of-the-art diffusion methods DDRM (20 NFEs) [25], DDNM (100 NFEs) [27], and DPS (1000 NFEs) [26]. CoModGAN, pscGAN, and rcGAN differ from pcaGAN only in generator regularization and CoModGAN's use of discriminator MBSD [55]. For rcGAN, DDNM, DDRM, and DPS, we use the authors' implementations from [45], [56], [57], and [58] with mask generation from [17]. FID and CFID were evaluated on our 20 000 image test set with $P = 1$.

**Results.** Table 4 shows test CFID, FID, LPIPS, and 40-sample generation time. The table shows that the proposed pcaGAN wins in CFID, FID and LPIPS, and that the four cGANs generate samples 3–4 orders-of-magnitude faster than DPS. Figure 4 shows five generated samples for each method under test, along with the true and masked image. pcaGAN shows better subjective quality than the competitors, as well as good diversity. Additional figures can be found in App. E.4.

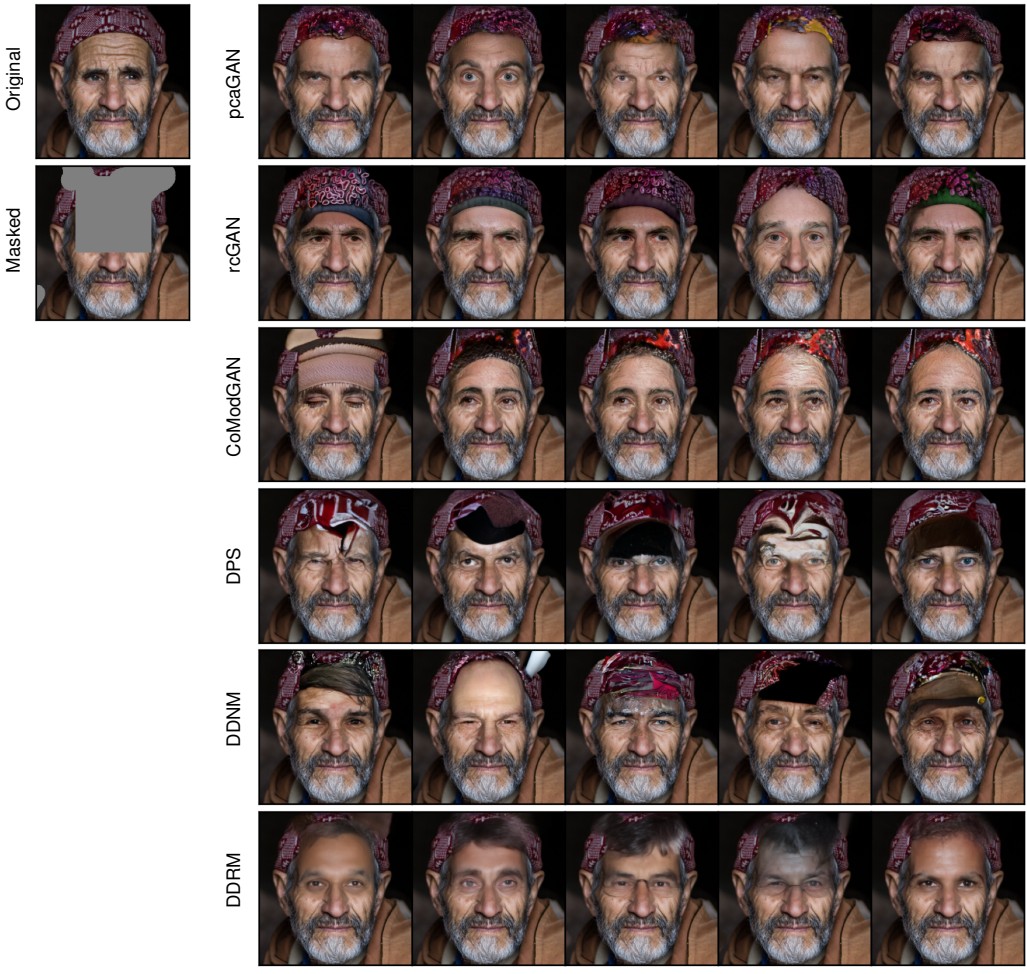

Figure 4: Example of inpainting a randomly generated mask on a $256 \times 256$ FFHQ face image.

## 5 Discussion

When training a cGAN, the overall goal is that the samples $\{\widehat{x}_i\}$ generated from a particular $y$ accurately represent the true posterior $p_{\mathsf{x}|\mathsf{y}}(\cdot|y)$. Achieving this goal is challenging when training from paired data $\{(x_t, y_t)\}$, because such datasets provide only one example of $x$ for each given $y$. Early methods like [15–17] focused on providing *some* variation among $\{\widehat{x}_i\}$, but did not aim for the correct variation. The rcGAN from [19] focused on providing the correct *amount* of variation by enforcing $\mathrm{tr}(\mathbf{\Sigma}_{\widehat{\mathsf{x}}|\mathsf{y}}) = \mathrm{tr}(\mathbf{\Sigma}_{\mathsf{x}|\mathsf{y}})$, and the proposed pcaGAN goes farther by encouraging $\mathbf{\Sigma}_{\widehat{\mathsf{x}}|\mathsf{y}}$ and $\mathbf{\Sigma}_{\mathsf{x}|\mathsf{y}}$ to agree along $K$ principal directions. Our experiments demonstrate that pcaGAN yields a notable improvement over rcGAN and outperforms contemporary diffusion approaches like DPS [26].

PCA principles have also been used in *unconditional* GANs, where the goal is to train a generator $G_{\boldsymbol{\theta}}$ that turns codes $z \sim \mathcal{N}(\mathbf{0}, \boldsymbol{I})$ into outputs $\widehat{x} = G_{\boldsymbol{\theta}}(z)$ that match the true marginal distribution $p_{\mathsf{x}}$ from which the training samples $\{x_t\}$ are drawn. For example, the eigenGAN from [59] aims to train in such a way that semantic attributes are learned (without supervision) and can be independently controlled by manipulating individual entries of $z$. But their goal is clearly different from ours.

**Limitations.** We acknowledge several limitations of our work. First, generating $P_{\mathsf{pca}} = 10K$ samples during training can impose a burden on memory when $x$ is high dimensional. In the multicoil MRI experiment, $x \in \mathbb{R}^d$ for $d = 2.4\mathrm{e}6$, which limited us to $K = 1$ at batch size 2. Second, although our focus is on designing a fast and accurate posterior sampler, more work is needed on how to best use the generated samples across different applications. Using them to compute rigorous uncertainty

intervals seems like a promising direction [60–62]. Third, the application to MRI is preliminary; additional tuning and validation is needed before it can be considered for clinical practice.

## 6  Conclusion

In this work, we proposed pcaGAN, a novel image-recovery cGAN that enforces correctness in the $K$ principal components of the conditional covariance matrix $\boldsymbol{\Sigma}_{\widehat{\mathsf{x}}|\mathsf{y}}$, as well as in the conditional mean $\boldsymbol{\mu}_{\widehat{\mathsf{x}}|\mathsf{y}}$ and trace-covariance $\operatorname{tr}(\boldsymbol{\Sigma}_{\widehat{\mathsf{x}}|\mathsf{y}})$. Experiments with synthetic Gaussian data showed pcaGAN outperforming both rcGAN [19] and NPPC [31] in Wasserstein-2 distance across a range of problem sizes. Experiments on MNIST denoising, accelerated multicoil MRI, and large-scale image inpainting showed pcaGAN outperforming several other cGANs and diffusion models in CFID, FID, PSNR, SSIM, LPIPS, and DISTS metrics. Furthermore, pcaGAN generates samples 3–4 orders-of-magnitude faster than the tested diffusion models. The proposed pcaGAN thus provides fast and accurate posterior sampling for image recovery problems, which enables uncertainty quantification, fairness in recovery, and easy navigation of the perception/distortion trade-off.

## Acknowledgments and Disclosure of Funding

The authors are funded in part by the National Institutes of Health under grant R01-EB029957.

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

# Supplementary Materials

## A  Synthetic Gaussian priors

Algorithm 2 captures how we construct the Gaussian priors used in Sec. 4.1.

---
**Algorithm 2** Gaussian prior generation

---
1: $\boldsymbol{\mu}_{\mathsf{x}}^{(100)} \sim \mathcal{N}(\boldsymbol{0}, \boldsymbol{I}_{100})$
2: $\widetilde{\lambda}_k^{(100)} \sim \mathcal{N}(0,1)$ for $k = 1, \ldots, 100$
3: $\lambda_k^{(100)} = |\widetilde{\lambda}_k^{(100)}|$ for $k = 1, \ldots, 100$
4: $\boldsymbol{u}_k^{(100)} \sim \mathcal{N}(\boldsymbol{0}, \boldsymbol{I}_{100})$ for $k = 1, \ldots, 100$
5: $[\boldsymbol{Q}, \boldsymbol{R}] = \mathrm{QRdecomp}([\boldsymbol{u}_1^{(100)} \boldsymbol{u}_2^{(100)} \ldots \boldsymbol{u}_{100}^{(100)}])$
6: $\boldsymbol{v}_k^{(100)} = [\boldsymbol{Q}]_{:k}$ for $k = 1, \ldots, 100$
7:
8: **for** $d = 90, 80, \ldots, 10$ **do**
9: $\quad \boldsymbol{\mu}_{\mathsf{x}}^{(d)} = [\boldsymbol{\mu}_{\mathsf{x}}^{(100)}]_{0:d}$
10: $\quad \lambda_k^{(d)} = \lambda_k^{(100)}$ for $k = 1, \ldots, d$
11: $\quad \boldsymbol{u}_k^{(d)} = \boldsymbol{v}_{k,0:d}^{(100)}$ for $k = 1, \ldots, d$
12: $\quad [\boldsymbol{Q}, \boldsymbol{R}] = \mathrm{QRdecomp}([\boldsymbol{u}_1^{(d)} \boldsymbol{u}_2^{(d)} \ldots \boldsymbol{u}_d^{(d)}])$
13: $\quad \boldsymbol{v}_k^{(d)} = [\boldsymbol{Q}]_{:k}$ for $k = 1, \ldots, d$
14: **end for**

---

We begin with dimension $d = 100$, generating random mean $\boldsymbol{\mu}_{\mathsf{x}}^{(100)} \in \mathbb{R}^{100}$ and eigenvalues $\{\lambda_k^{(100)}\}_{k=1}^{100}$. To construct the eigenvectors $\{\boldsymbol{v}_k^{(100)}\}_{k=1}^{100}$, we perform a QR decomposition on a $100 \times 100$ matrix with i.i.d. $\mathcal{N}(0,1)$ entries and set $\boldsymbol{v}_k^{(100)}$ as the $k$th column of $\boldsymbol{Q}$. For each remaining $d \in \{90, 80, \ldots, 10\}$, we construct $\boldsymbol{\mu}_{\mathsf{x}}^{(d)}$, $\{\lambda_k^{(d)}\}$, and $\boldsymbol{u}_k^{(d)}$ by truncating the previous quantities to ensure some level of continuity across $d$.

## B  Conditional Fréchet inception distance

We quantify posterior-approximation accuracy using conditional Fréchet inception distance (CFID) [38], which approximates the conditional Wasserstein distance

$$\mathrm{CWD} \triangleq \mathrm{E}_{\mathsf{y}}\{W_2(p_{\mathsf{x}|\mathsf{y}}(\cdot, \boldsymbol{y}), p_{\widehat{\mathsf{x}}|\mathsf{y}}(\cdot, \boldsymbol{y}))\}. \tag{B.1}$$

In (B.1), $W_2(p_{\mathsf{a}}, p_{\mathsf{b}})$ denotes the Wasserstein-2 distance between distributions $p_{\mathsf{a}}$ and $p_{\mathsf{b}}$, defined as

$$W_2(p_{\mathsf{a}}, p_{\mathsf{b}}) \triangleq \min_{p_{\mathsf{a},\mathsf{b}} \in \Pi(p_{\mathsf{a}}, p_{\mathsf{b}})} \mathrm{E}_{\mathsf{a},\mathsf{b}}\{\|\boldsymbol{a} - \boldsymbol{b}\|_2^2\}, \tag{B.2}$$

where $\Pi(p_{\mathsf{a}}, p_{\mathsf{b}}) \triangleq \{p_{\mathsf{a},\mathsf{b}} : p_{\mathsf{a}} = \int p_{\mathsf{a},\mathsf{b}} \, \mathrm{d}\boldsymbol{b} \text{ and } p_{\mathsf{b}} = \int p_{\mathsf{a},\mathsf{b}} \, \mathrm{d}\boldsymbol{a}\}$ denotes the set of joint distributions $p_{\mathsf{a},\mathsf{b}}$ with marginal distributions $p_{\mathsf{a}}$ and $p_{\mathsf{b}}$. Similarly to how FID [39] is computed for marginal distributions, CFID approximates (B.1) for conditional distributions. In particular, the random vectors $\boldsymbol{x}$, $\widehat{\boldsymbol{x}}$, and $\boldsymbol{y}$ are replaced by low-dimensional embeddings $\underline{\boldsymbol{x}}$, $\underline{\widehat{\boldsymbol{x}}}$, and $\underline{\boldsymbol{y}}$, generated by the convolutional layers of a deep network (e.g., InceptionV3 or VGG-16 or AlexNet), and the embedding distributions $p_{\underline{\mathsf{x}}|\underline{\mathsf{y}}}$ and $p_{\underline{\widehat{\mathsf{x}}}|\underline{\mathsf{y}}}$ are approximated by multivariate Gaussians. We compute CFID using the code from [45], which is described in [19].

## C  Additional MRI results for acceleration $R = 4$

Table 5 shows PSNR, SSIM, LPIPS [50], and DISTS [51] for the $P$-sample average $\widehat{\boldsymbol{x}}_{(P)}$ at $P \in \{1, 2, 4, 8, 16, 32\}$ for $R = 4$. In this case, the E2E-VarNet attains the best PSNR and SSIM, while pcaGAN performs best in LPIPS and DISTS when $P = 2$ and $P = 1$, respectively. These results,

Table 5: Average PSNR, SSIM, LPIPS, and DISTS of $\widehat{x}_{(P)}$ versus $P$ for $R = 4$ MRI

| | PSNR↑ | | | | | | SSIM↑ | | | | | |
| Model | $P\!=\!1$ | $P\!=\!2$ | $P\!=\!4$ | $P\!=\!8$ | $P\!=\!16$ | $P\!=\!32$ | $P\!=\!1$ | $P\!=\!2$ | $P\!=\!4$ | $P\!=\!8$ | $P\!=\!16$ | $P\!=\!32$ |
|---|---|---|---|---|---|---|---|---|---|---|---|---|
| E2E-VarNet (Sriram et al. [43]) | **39.93** | - | - | - | - | - | **0.9641** | - | - | - | - | - |
| Langevin (Jalal et al. [22]) | 36.04 | 37.02 | 37.65 | 37.99 | 38.17 | 38.27 | 0.8989 | 0.9138 | 0.9218 | 0.9260 | 0.9281 | 0.9292 |
| cGAN (Adler & Öktem [16]) | 35.63 | 36.64 | 37.24 | 37.56 | 37.73 | 37.82 | 0.9330 | 0.9445 | 0.9478 | 0.9480 | 0.9477 | 0.9473 |
| pscGAN (Ohayon et al. [29]) | 39.44 | 39.46 | 39.46 | 39.47 | 39.47 | 39.47 | 0.9558 | 0.9546 | 0.9539 | 0.9535 | 0.9533 | 0.9532 |
| rcGAN (Bendel et al. [19]) | 36.96 | 38.14 | 38.84 | 39.24 | 39.44 | 39.55 | 0.9440 | 0.9526 | 0.9544 | 0.9542 | 0.9537 | 0.9533 |
| pcaGAN (Ours) | 37.32 | 38.43 | 39.11 | 39.47 | 39.67 | 39.77 | 0.9463 | 0.9541 | 0.9557 | 0.9553 | 0.9546 | 0.9542 |
| | LPIPS↓ | | | | | | DISTS↓ | | | | | |
| Model | $P\!=\!1$ | $P\!=\!2$ | $P\!=\!4$ | $P\!=\!8$ | $P\!=\!16$ | $P\!=\!32$ | $P\!=\!1$ | $P\!=\!2$ | $P\!=\!4$ | $P\!=\!8$ | $P\!=\!16$ | $P\!=\!32$ |
| E2E-VarNet (Sriram et al. [43]) | 0.0316 | - | - | - | - | - | 0.0859 | - | - | - | - | - |
| Langevin (Jalal et al. [22]) | 0.0545 | 0.0394 | 0.0336 | 0.0320 | 0.0317 | 0.0316 | 0.1116 | 0.0921 | 0.0828 | 0.0793 | 0.0781 | 0.0777 |
| cGAN (Adler & Öktem [16]) | 0.0285 | 0.0255 | 0.0273 | 0.0298 | 0.0316 | 0.0327 | 0.0972 | 0.0857 | 0.0878 | 0.0930 | 0.0967 | 0.0990 |
| pscGAN (Ohayon et al. [29]) | 0.0245 | 0.0247 | 0.0248 | 0.0249 | 0.0249 | 0.0249 | 0.0767 | 0.0790 | 0.0801 | 0.0807 | 0.0810 | 0.0811 |
| rcGAN (Bendel et al. [19]) | 0.0175 | 0.0164 | 0.0188 | 0.0216 | 0.0235 | 0.0245 | 0.0546 | 0.0563 | 0.0667 | 0.0755 | 0.0809 | 0.0837 |
| pcaGAN (Ours) | 0.0164 | **0.0159** | 0.0188 | 0.0214 | 0.0231 | 0.0242 | **0.0542** | 0.0548 | 0.0662 | 0.0754 | 0.0811 | 0.0843 |

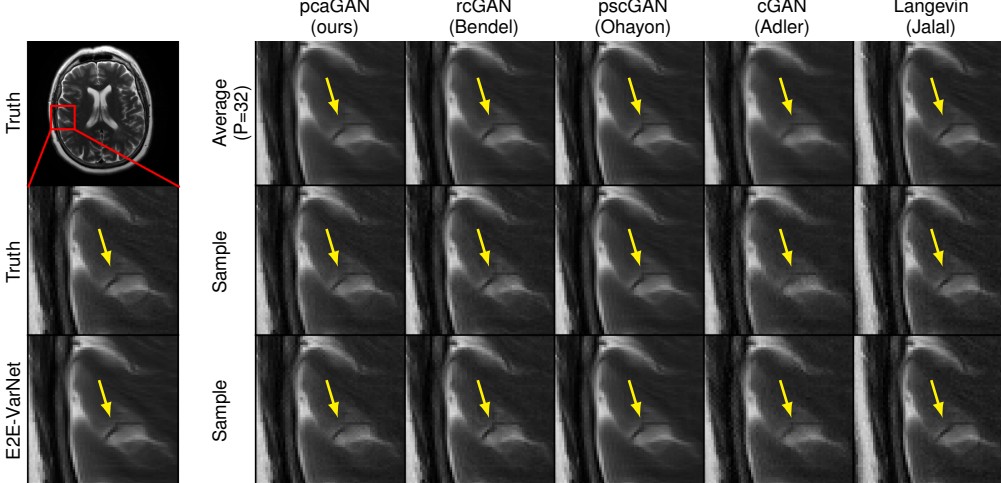

Figure C.1: Example MRI recoveries at $R = 4$. Arrows highlight meaningful variations.

in conjunction with the $R = 8$ results discussed in Sec. 4.3, show that pcaGAN yields a notable improvement over rcGAN in all metrics.

Figure C.1 shows zoomed versions of two recoveries $\widehat{x}_i$ and the sample average $\widehat{x}_{(P)}$ with $P = 32$.

# D   Implementation details

In each experiment, all cGANs were trained using the Adam optimizer with a learning rate of $10^{-3}$, $\beta_1 = 0$, and $\beta_2 = 0.99$ as in [55].

## D.1   Synthetic Gaussian recovery

**cGAN training.** We choose $\beta_{\mathsf{adv}} = 10^{-5}$, $n_{\mathsf{batch}} = 64$, $P_{\mathsf{rc}} = 2$, and train for 100 epochs for both rcGAN and pcaGAN. Running PyTorch on a server with 4 Tesla A100 GPUs, each with 82 GB of memory, the cGAN training for $d = 100$ takes approximately 8 hours, with training time decreasing with smaller $d$. For pcaGAN, we choose $K = d$ for each $d$ in this experiment (unless otherwise noted) and $\beta_{\mathsf{pca}} = 10^{-2}$.

**cGAN architecture.** We exploit the Gaussian nature of the problem, constructing $G_{\boldsymbol{\theta}}$ with two dense layers; one which takes in $\boldsymbol{y}$ as input and one which takes in $\boldsymbol{z}$ as input. The output of each layer is added, yielding $\widehat{\boldsymbol{x}}$. Similarly, $D_{\boldsymbol{\phi}}$ is comprised of a single dense layer which takes in the concatenation of $\boldsymbol{x}$ / $\widehat{\boldsymbol{x}}$ and $\boldsymbol{y}$ and outputs a scalar. We use this architecture for both rcGAN and pcaGAN. Note that there is no listed license for rcGAN.

**NPPC.** For NPPC, we use the suggested hyperparameters from [31] and opt to train the MMSE network *before* training NPPC. We use the suggested architectures from their Gaussian denoising experiment and train for 100 epochs with $n_{\mathsf{batch}} = 64$. We leverage the authors' implementation in [36], modifying it for this Gaussian problem. There is no listed license for NPPC.

## D.2 MNIST denoising

The MNIST dataset is available under the GNU general public license, which we respect through our use.

**cGAN training.** We choose $\beta_{\mathsf{adv}} = 10^{-5}$, $n_{\mathsf{batch}} = 64$, $P_{\mathsf{rc}} = 2$, and train for 125 epochs for both rcGAN and pcaGAN. Running PyTorch on a server with 4 Tesla A100 GPUs, each with 82 GB of memory, the cGAN training for $d = 100$ takes approximately 8 hours, with training time decreasing with smaller $d$. For pcaGAN, we train two models, one with $K = 5$ and one with $K = 10$. In both cases, $\beta_{\mathsf{pca}} = 10^{-1}$, $E_{\mathsf{evec}} = 25$, and $E_{\mathsf{eval}} = 50$.

**cGAN architecture.** For both cGANs, the generator is the standard UNet which takes in the concatenation of $\boldsymbol{y}$ and code $\boldsymbol{z}$. The network consists of 3 pooling layers and 32 initial channels. The convolutions use a kernel of size $3 \times 3$, instance normalization, and leaky ReLU activations with a negative slope of $0.1$. For the encoder portion of the UNet, we use max pooling with a kernel size of $2 \times 2$ to reduce spatial resolution by a factor of 2. For the decoder portion of the UNet, we use nearest-neighbor interpolation to increase spatial resolution by a factor of 2. The discriminator is simply the encoder portion of the UNet with an additional dense layer appended to map the UNet's latent space to a scalar output. Note that there is no listed license for rcGAN.

**NPPC.** For NPPC, we do not modify the authors' implementation in [36] in any way. We first train the MMSE reconstruction network and then train the NPPC network. There is no listed license for NPPC.

## D.3 Accelerated MRI

For our MRI experiments, we use the fastMRI dataset which is available under the MIT license, which we respect through our use.

**cGAN training.** We choose $\beta_{\mathsf{adv}} = 10^{-5}$, $\beta_{\mathsf{pca}} = 10^{-2}$, $n_{\mathsf{batch}} = 2$, $P_{\mathsf{rc}} = 2$, $K = 1$, $E_{\mathsf{evec}} = 25$, and $E_{\mathsf{eval}} = 50$ for pcaGAN. For rcGAN, pscGAN , and Adler's cGAN, we use the hyperparameters and training procedure described in [19]. All models were trained for 100 epochs using the Adam optimizer [35] with a learning rate of $10^{-3}$, $\beta_1 = 0$, and $\beta_2 = 0.99$, as in [55]. Running PyTorch on a server with 4 Tesla A100 GPUs, each with 82 GB of memory, the training of each MRI cGAN took approximately 1 day. Note that there is no listed license for rcGAN, pscGAN, or Adler and Öktem's cGAN.

**cGAN architecture.** All four cGANs used the same generator and discriminator architectures described in [19], except that Adler and Öktem's discriminator used extra input channels to facilitate the 3-input loss.

**E2E-VarNet.** For the Sriram et al.'s E2E-VarNet [43], we use the same training procedure and hyperparameters outlined in [22] with modification to the sampling pattern as in [19]. As in [22], we use the SENSE-based coil-combined image as the ground truth instead of the RSS image. The E2E-VarNet is available under the MIT license, which we respect.

**Langevin approach.** For Jalal et al.'s MRI approach [22], we do not modify the authors' implementation from [46] other than replacing the default sampling pattern with the GRO sampling mask. We borrow both generated samples and results from [19]. The authors' code is available under the MIT license, which we respect.

## D.4 FFHQ Inpainting

For our inpainting experiment, we use the FFHQ dataset which is available under the Creative Commons BY-NC-SA 4.0 license, which we respect through our use.

**cGAN training.** We choose $\beta_{\mathsf{adv}} = 5 \times 10^{-3}$, $\beta_{\mathsf{pca}} = 10^{-3}$, $n_{\mathsf{batch}} = 5$, $P_{\mathsf{rc}} = 2$, $K = 2$, $E_{\mathsf{evec}} = 25$, and $E_{\mathsf{eval}} = 50$ for pcaGAN. Running PyTorch on a server with 4 Tesla A100 GPUs, each with 82 GB of memory, the training of our cGAN took approximately 1.5 days.

**cGAN architecture.** As in [19], we use the CoModGAN networks from [17] which extend the StyleGAN2 [34] network. The StyleGAN2 architecture is available under the NVIDIA Source Code License, which we respect.

**rcGAN.** We follow the training procedure outlined in [19], only modifying the inpainting mask to be random. The total training time on a server with 4 NVIDIA A100 GPUs, each with 82 GB of memory, is roughly 1 day. There is no listed license for rcGAN.

**pscGAN.** We use the same training procedure outlined in [19], modifying the inpainting masks to be random and using the $\mathcal{L}_{2,P}$ objective described briefly in Sec. 2 with $P = 8$. The total training time on a server with 4 NVIDIA A100 GPUs, each with 82 GB of memory, is roughly 1.5 days. There is no listed license for pscGAN.

**CoModGAN.** We use the PyTorch implementation of CoModGAN from [63] and train the model. The total training time on a server with 4 NVIDIA A100 GPUs, each with 82 GB of memory, is roughly 1 day. There is no listed license for CoModGAN, beyond the NVIDIA Source Code License.

### D.4.1 Diffusion Methods

For all three diffusion methods, we use the pretrained weights from [26].

**DPS.** We use the suggested settings for the $256 \times 256$ FFHQ dataset and the code from the official PyTorch implementation [58]. We found the LPIPS-minimizing step-size $\zeta$ via grid search over a 1000 image validation set. We generate 1 sample for all 20 000 images in our test set, using a batch-size of 1 and 1000 NFEs. The total generation time on a server with 4 NVIDIA A100 GPUs, each with 82 GB of memory, is roughly 9 days. There is no listed license for DPS.

**DDNM.** We use the code from the official PyTorch implementation [56]. We generate 1 sample for all 20 000 images in our test set, using a batch-size of 1 and 100 NFEs. The total generation time on a server with 4 NVIDIA A100 GPUs, each with 82 GB of memory, is roughly 1.5 days. There is no listed license for DDNM.

**DDRM.** We use the code from the official PyTorch implementation [57]. We generate 1 sample for all 20 000 images in our test set, using a batch-size of 1 and 20 NFEs. The total generation time on a server with 4 NVIDIA A100 GPUs, each with 82 GB of memory, is roughly 5.5 hours. There is no listed license for DDRM.

# E  Additional reconstruction plots

## E.1  MNIST denoising

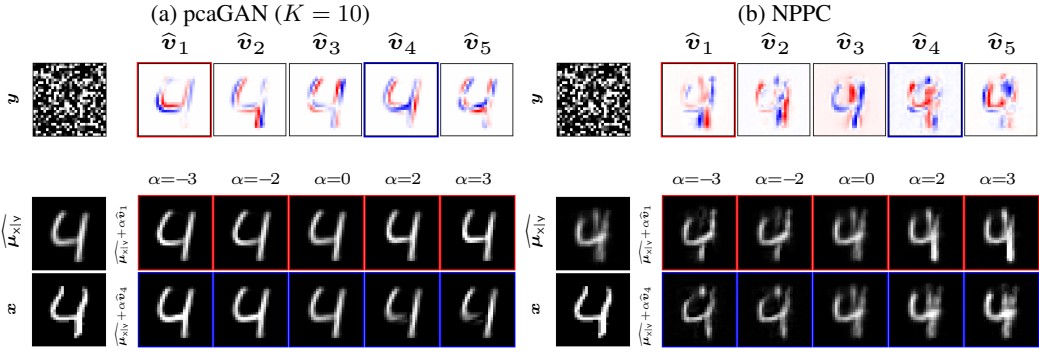

Figure E.1: For (a) pcaGAN and (b) NPPC, this figure shows the true image $x$, noisy measurements $y$, the conditional mean $\widehat{\mu_{x|y}}$, principal eigenvectors $\{\widehat{v}_k\}$, and two perturbations of $\widehat{\mu_{x|y}}$.

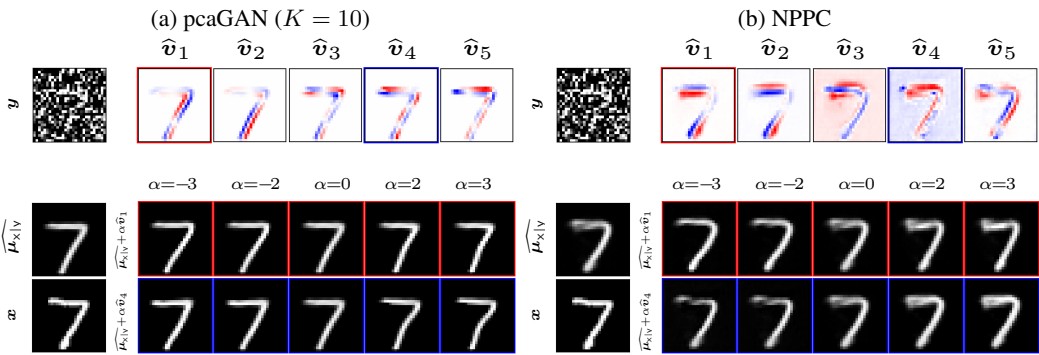

Figure E.2: For (a) pcaGAN and (b) NPPC, this figure shows the true image $x$, noisy measurements $y$, the conditional mean $\widehat{\mu_{x|y}}$, principal eigenvectors $\{\widehat{v}_k\}$, and two perturbations of $\widehat{\mu_{x|y}}$.

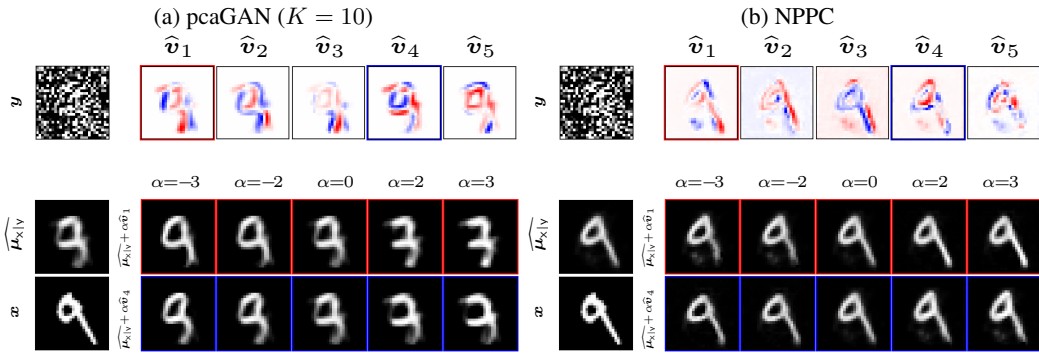

Figure E.3: For (a) pcaGAN and (b) NPPC, this figure shows the true image $x$, noisy measurements $y$, the conditional mean $\widehat{\mu_{x|y}}$, principal eigenvectors $\{\widehat{v}_k\}$, and two perturbations of $\widehat{\mu_{x|y}}$.

## E.2 MRI at acceleration $R = 4$

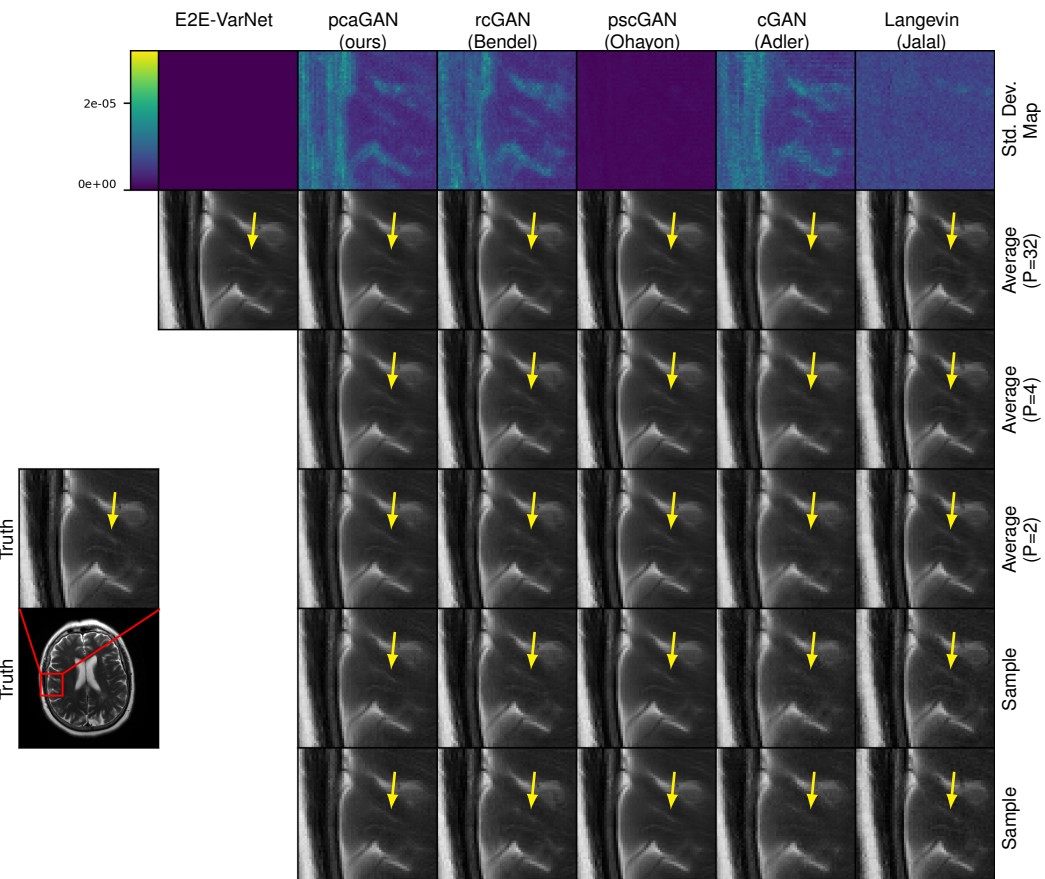

Figure E.4: Example $R = 4$ MRI reconstruction. Row one: pixel-wise SD with $P = 32$, Row two: $\widehat{\boldsymbol{x}}_{(P)}$ with $P = 32$, Row three: $\widehat{\boldsymbol{x}}_{(P)}$ with $P = 4$, Row four: $\widehat{\boldsymbol{x}}_{(P)}$ with $P = 2$, Rows five and six: posterior samples. The arrows indicate regions of meaningful variation across posterior samples.

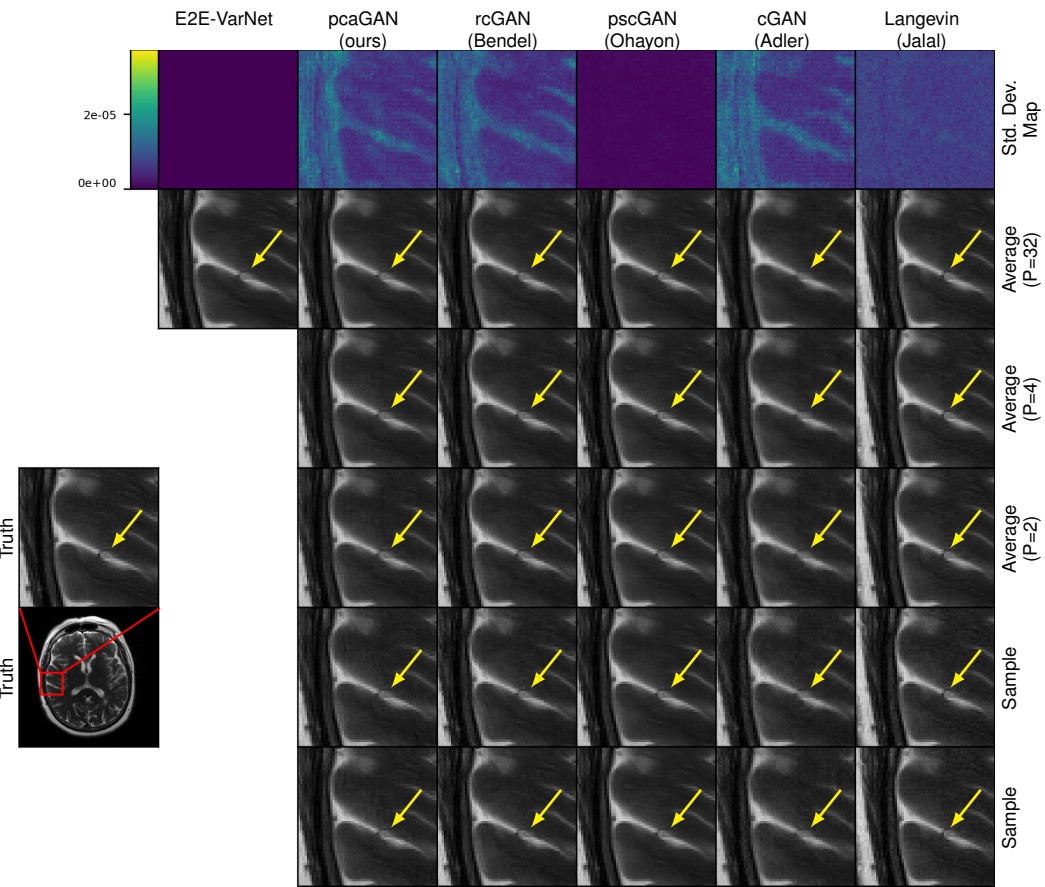

Figure E.5: Example $R = 4$ MRI reconstruction. Row one: pixel-wise SD with $P = 32$, Row two: $\widehat{\boldsymbol{x}}_{(P)}$ with $P = 32$, Row three: $\widehat{\boldsymbol{x}}_{(P)}$ with $P = 4$, Row four: $\widehat{\boldsymbol{x}}_{(P)}$ with $P = 2$, Rows five and six: posterior samples. The arrows indicate regions of meaningful variation across posterior samples.

## E.3 MRI at acceleration $R = 8$

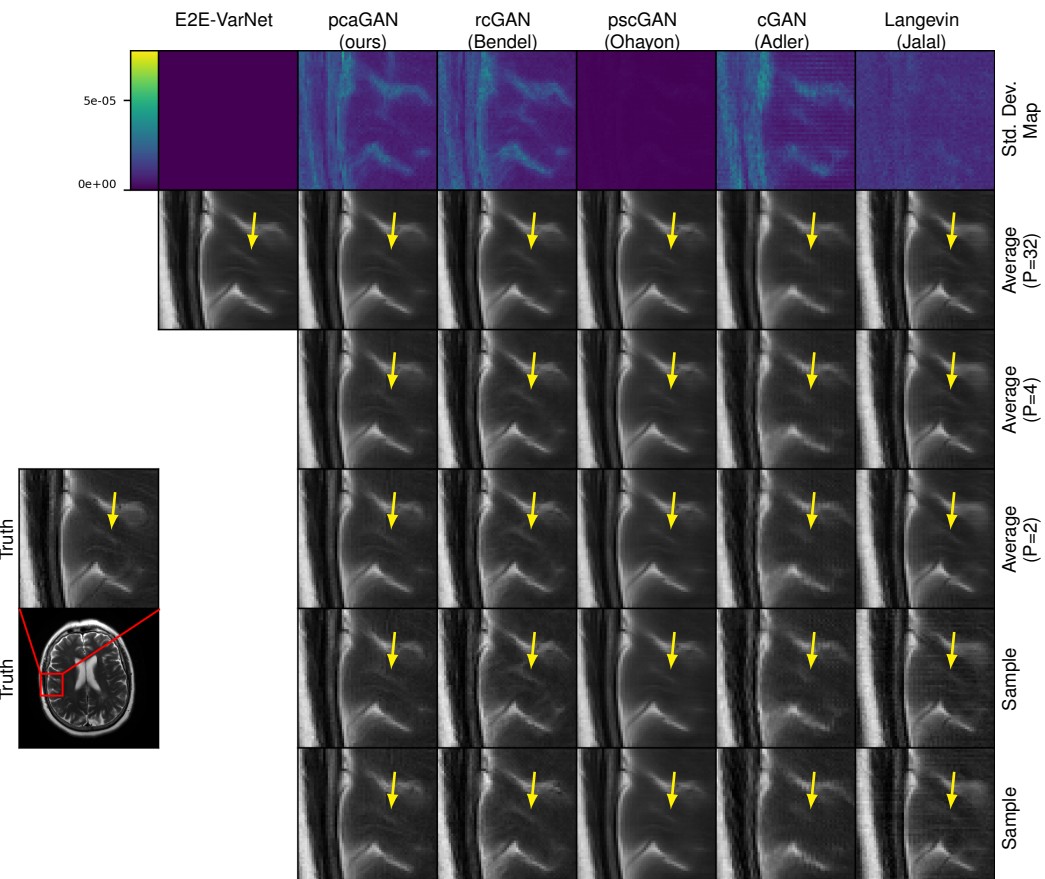

Figure E.6: Example $R = 8$ MRI reconstruction. Row one: pixel-wise SD with $P = 32$, Row two: $\widehat{x}_{(P)}$ with $P = 32$, Row three: $\widehat{x}_{(P)}$ with $P = 4$, Row four: $\widehat{x}_{(P)}$ with $P = 2$, Rows five and six: posterior samples. The arrows indicate regions of meaningful variation across posterior samples.

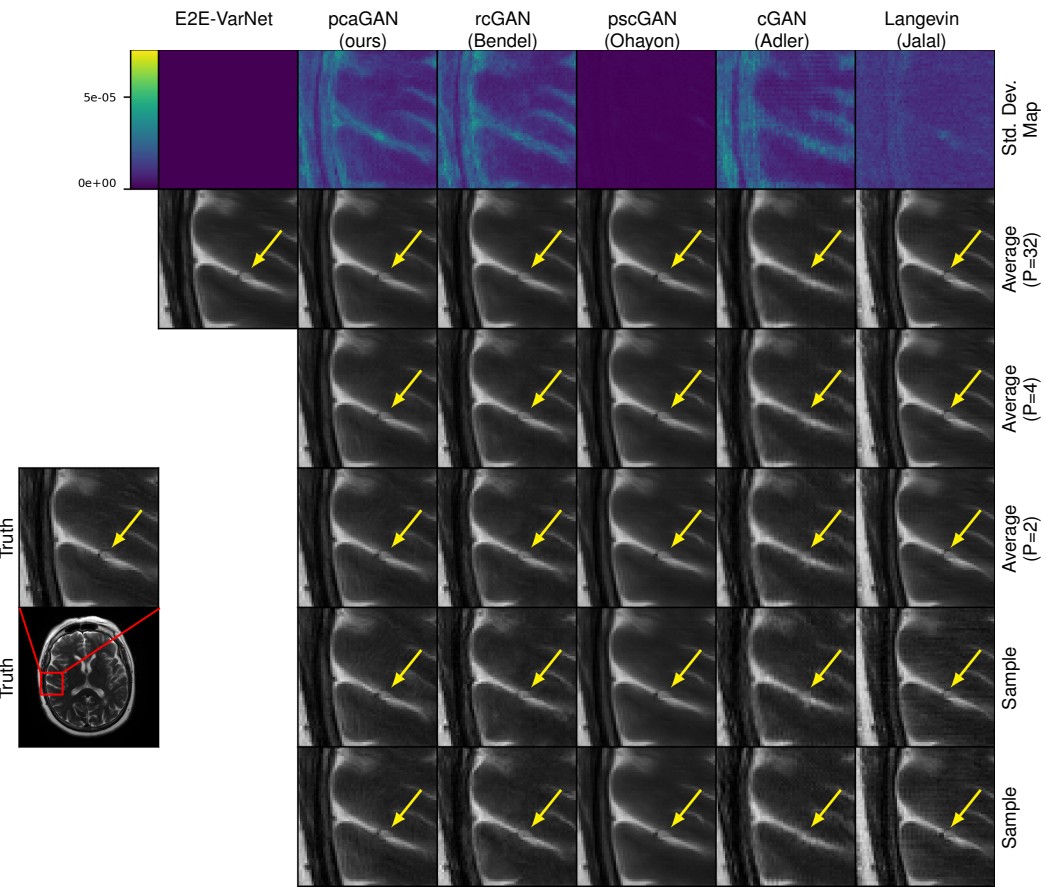

Figure E.7: Example $R = 8$ MRI reconstruction. Row one: pixel-wise SD with $P = 32$, Row two: $\widehat{\boldsymbol{x}}_{(P)}$ with $P = 32$, Row three: $\widehat{\boldsymbol{x}}_{(P)}$ with $P = 4$, Row four: $\widehat{\boldsymbol{x}}_{(P)}$ with $P = 2$, Rows five and six: posterior samples. The arrows indicate regions of meaningful variation across posterior samples.

## E.4 Inpainting

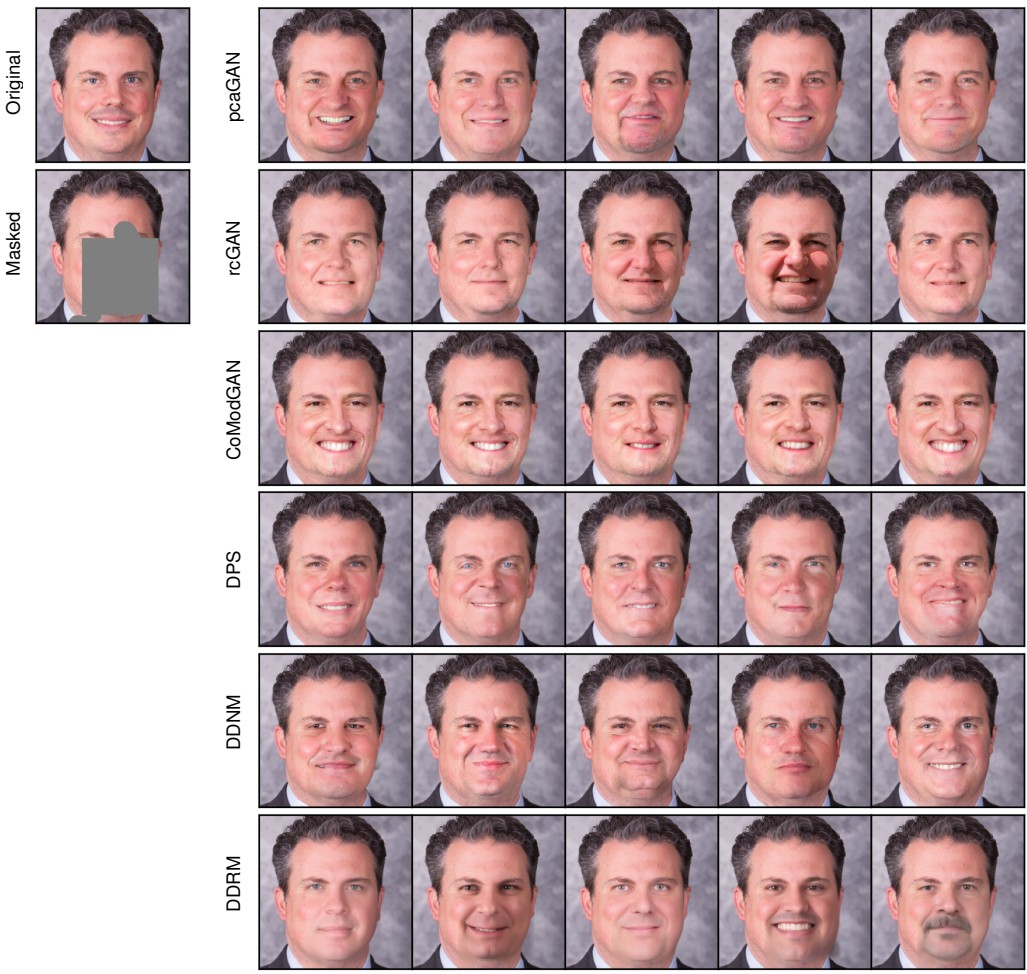

Figure E.8: Example of inpainting a randomly generated mask on a $256 \times 256$ FFHQ face image.

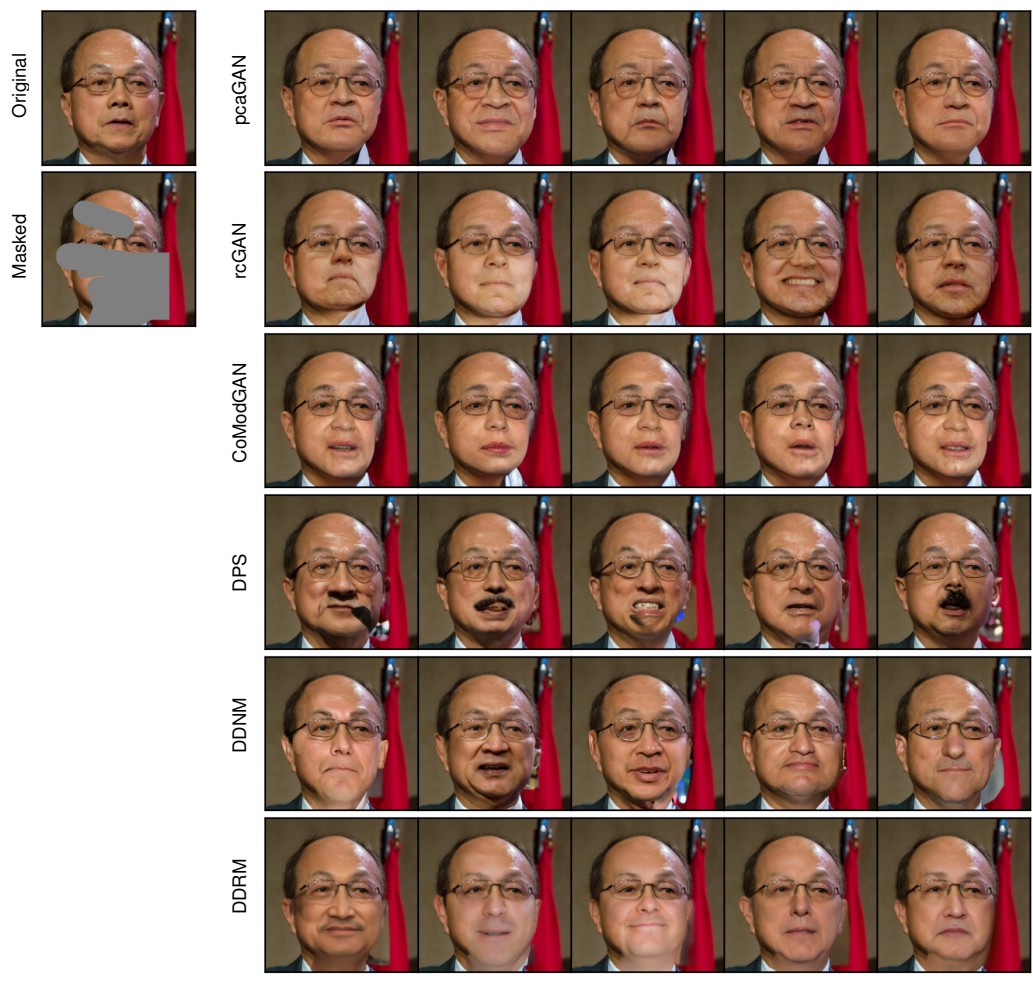

Figure E.9: Example of inpainting a randomly generated mask on a $256 \times 256$ FFHQ face image.

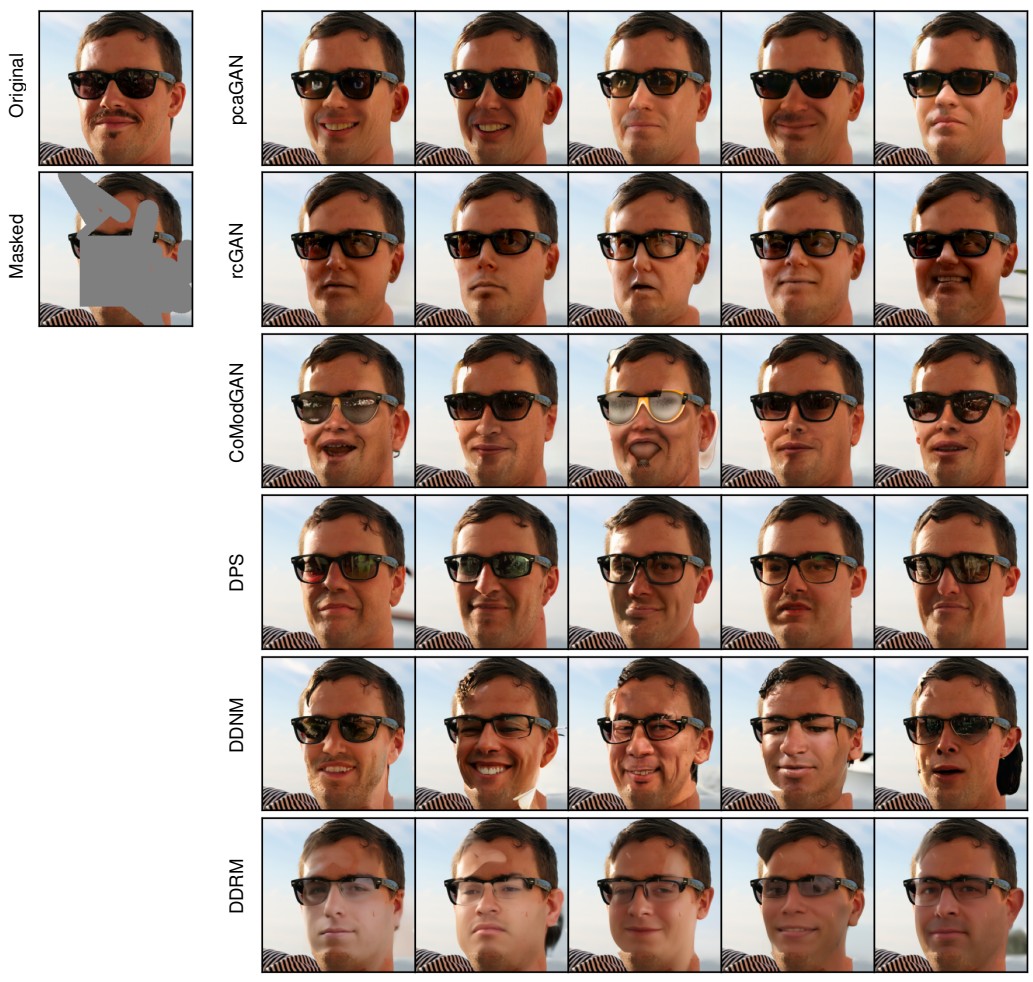

Figure E.10: Example of inpainting a randomly generated mask on a $256 \times 256$ FFHQ face image.

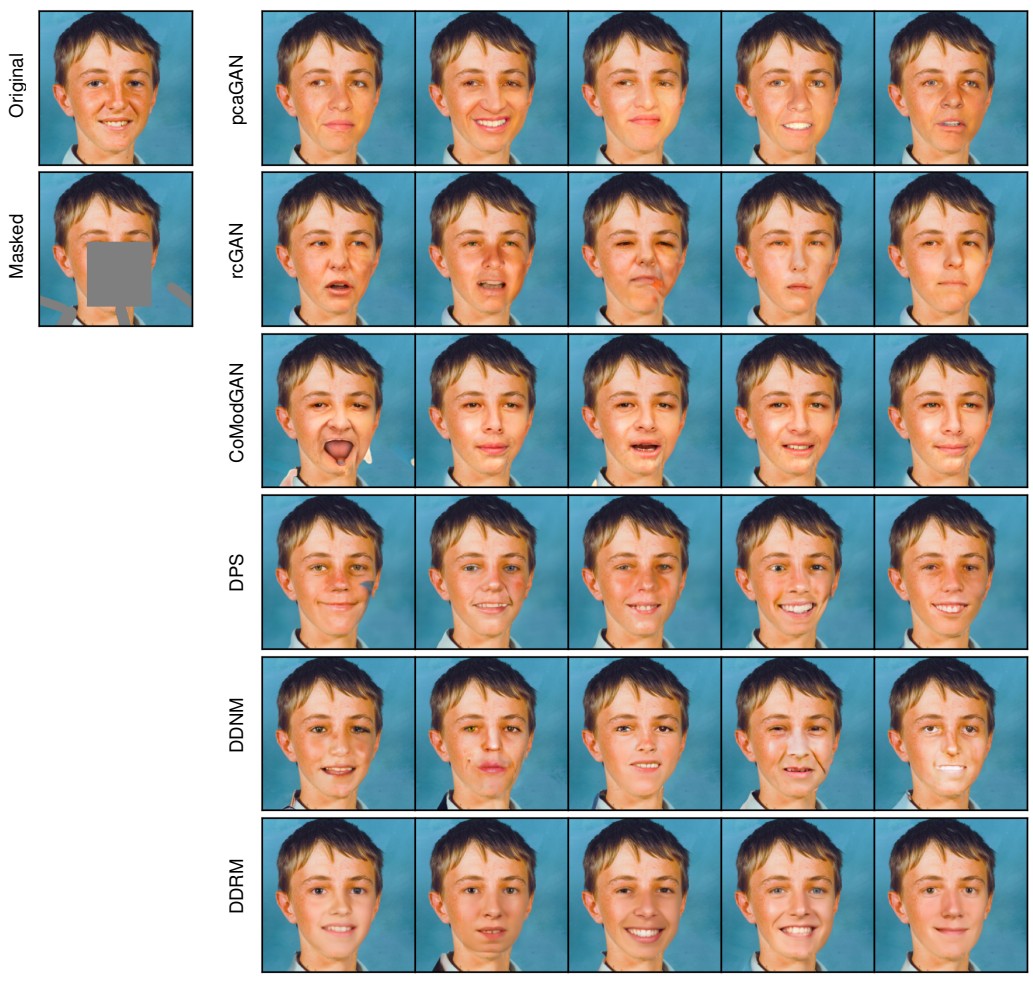

Figure E.11: Example of inpainting a randomly generated mask on a $256 \times 256$ FFHQ face image.

