# OpenReview forum: "pcaGAN: Improving Posterior-Sampling cGANs via Principal Component Regularization"
_NeurIPS.cc/2024/Conference — NeurIPS 2024 poster_

### Official Review · Reviewer_cM4t · 2024-06-12

**Soundness:** 2
**Presentation:** 1
**Contribution:** 3
**Rating:** 4
**Confidence:** 5

**Summary:**

To solve image inverse problems (denoising, super resolution, etc.), many works attempt to sample from the posterior distribution of ground truth images given a degraded measurement. One line of works propose to train a stochastic neural network as a CGAN (conditioned on the degraded image and an additional random seed as inputs), while regularizing the conditional mean and the conditional covariance matrix of the outputs. This paper proposes yet another, improved regularization term for such a CGAN approach. Specifically, the authors propose to correct (regularize) the K principal components of the conditional covariance matrix in additional to the conditional mean and the trace of the covariance.

**Strengths:**

The paper is mostly written well. The proposed method is demonstrated on toy examples (such as Gaussian random signals and MNIST images), which I find appealing to gain intuition into the advantages of the proposed approach. The paper shows results on MRI reconstruction, which is not very common to report but an incredibly important inverse problem to solve.

**Weaknesses:**

Even though I personally like this type of work and the proposed approach, unfortunately I see several major weaknesses in this paper, starting even with wrong/inaccurate claims in the introduction & motivation. I'll list some of the weaknesses below.

1) I disagree with most of the claims made in L17 onwards regarding the shortcomings of point estimators (introduction). The perception-distortion tradeoff theorem in [6] does not prove anything specific about point estimators (a claim made in L20-L21). In fact, this theorem proves that distortion can only be at odds with perceptual quality (but now always), regardless of whether the estimator is "deterministic" or not. Moreover, [6] defines perceptual quality as the statistical distance (e.g., KL) between $p_{\hat{X}}$ (the distribution of the reconstructions) and $p_{X}$ (the distribution of the source images), and not as the distance between $\hat{x}$ and the manifold of natural images (as claimed in L22). What is the definition of such a distance? Lastly, the perception-distortion tradeoff is not always a strict one, as hinted in L21. This tradeoff  depends on the distortion measure, as discussed by the authors of [6]. Some distortion measures (e.g., "perceptual" losses in the deep image features space) virtually do not contradict with perceptual quality (as demonstrated experimentally in [6]). I would be pleased to see a complete revision of this paragraph in which the errors are corrected, or otherwise a clear explanation for why am I wrong. This is the entry point of the paper, it should at least be accurate.

2) I also do not agree with L27: "to address these shortcomings, posterior-sampling-based image recovery has been proposed". Posterior sampling as a solution for inverse problems has been proposed and suggested, e.g., with Langevin dynamics (though, it didn't work properly for high-dimensional data) much before we knew anything about the perception-distortion tradeoff or fairness issues in image reconstruction. As I understand it, the true motivation for the rise of posterior-sampling is the following: We are often dealing with ill-posed inverse problems, and performing posterior sampling allows us to conduct uncertainty quantification of the inverse task at hand, explore the space of solutions visually, and make more informed decisions in down-stream tasks. It has also been shown that stochastic estimators are more robust to adversarial attacks [1]. I don't see why the rise of posterior sampling is linked to the perception-distortion tradeoff and/or to fairness. Please provide evidence (citations and explanation) if that's the case.

3) L30: "posterior samples will display attributes that reflect minority and majority populations in the training data". As far as I know, this is often incorrect. In fact, we may often require to produce an intractable amount of posterior samples to observe an outcome for a minority. Please see [2] as an example.

4) L43: You are missing citations (and comparison) with highly related work that proposed to use posterior sampling using a CGAN with conditional mean and conditional covariance regularizations, prior to the work of Bendel et al. For example, [3] and [4]. In fact, the work of Bendel et al. is based on the idea proposed in [2] (to regularize the neural network by enforcing agreement of the average posterior sample with the posterior mean), and in [3] (to use a second-moment penalty of the covariance matrix as well), as you suggest in L75. As far as I understand, your algorithm is a different (and maybe improved) version of [4], but I don't know which one is better since a comparison with [3] and [4] is not reported in the paper. Please address the differences in your work so I could better understand why it's not appropriate to cite such papers and/or include them in your analysis.

5) I would suggest explaining L76-L79 better, by adding appropriate definitions for all the terminology used. For instance, the definition of a $P_{rc}$-sample supervised-$l_{1}$ loss is missing and unclear. I would be happy to see a more self-contained background section. Make sure every notation you are using is clearly defined before using it.

6) Some results seem wrong to me. For example, the results of DPS in Figure 4. I have been using DPS in many different image inverse problems, and never observed such inconsistencies between the outputs and the inputs (see, e.g., Figure 14 in the DPS paper appendices, which shows almost perfect consistency with the measurements for severe inpainting tasks). The FID results of DPS also seem odd to me. I find it very hard to believe that a CGAN algorithm would outperform DPS, given that the hyper-parameters of DPS are properly tuned. Can you please provide the code you used to execute these experiments so I could verify them? Maybe the hyperparemeters you used to run DPS are suboptimal?

7) The contribution of this work is quite limited in my opinion. The proposed algorithm only slightly improves its predecessor (Badler et al.), and the experimental evaluation is limited. A comparison with more works is missing, such as [3] and [4], and other, more recent state-of-the-art stochastic image restoration methods that produce better results than DPS. What about DDNM, DDRM, etc.? Why are these not included in the analysis? (besides the fact that they don't "promise" posterior sampling). I would be happy if you could clarify the contribution of your work and its impact on the field, as it seems quite incremental to me.


[1] Reasons for the Superiority of Stochastic Estimators over Deterministic Ones: Robustness, Consistency and Perceptual Quality, Guy Ohayon et al., ICML 2023

[2] From Posterior Sampling to Meaningful Diversity in Image Restoration, Noa Cohen et al, ICLR 2024.

[3] High Perceptual Quality Image Denoising with a Posterior Sampling CGAN, Guy Ohayon et al., ICCV workshops 2021.

[4] High-Perceptual Quality JPEG Decoding via Posterior Sampling, Sean Man et al, CVPR workshops 2023.

**Questions:**

Please address each of the weaknesses.

**Limitations:**

The authors do provide a dedicated limitations section, but I observe additional limitations in this paper which are not addressed, such as incremental novelty of the proposed algorithm over previous work, limited experimental section (comparing only to one predecessor), and more, as explained in the weaknesses section.

---

> ### Author Rebuttal · Authors · 2024-08-06
>
> We sincerely thank the reviewer for their valuable feedback and time spent reviewing.
>
> - **Reviewer**: I disagree with most of the claims made regarding the shortcomings of point estimators. The perception-distortion (PD) tradeoff theorem in [6] does not prove anything specific about point estimators.
>
>     **Response**: We'll try to explain again.  We'll refer to the full paper
>     >    [PD] https://arxiv.org/pdf/1711.06077v4
>
>     not the IEEE version [6] missing figures and appendices.
>
>     [PD] considers the setting in which a *single* (possibly random) image estimate $\hat{X}(y)$ is constructed from a given measurement $y$.
>     Examples in [PD] include the conditional-mean estimate $\hat{X}(y)=E\\{X|Y=y\\}$, which achieves optimal MSE distortion but suboptimal perceptual quality, and the "conditional draw" estimator, where $\hat{X}(y)$ is a single random draw from the posterior $p(X|Y=y)$, which achieves perfect perceptual quality but suboptimal distortion (see Thm.3 and Fig.7 in [PD]).
>     The key takeaway from the PD tradeoff is that no $\hat{X}(y)$ can *simultaneously* achieve optimal distortion and perfect perceptual quality.
>     To achieve $K$ points along the tradeoff, [PD] proposed to train a GAN $K$ times, with a different loss each time.
>
>     Our point is that by drawing *many* posterior samples $\\{\hat{X}\_i(y)\\}\_{i=1}^P$, where $\hat{X}\_i(y)\sim$ iid $p(X|Y=y)$, one circumvents the PD tradeoff in the sense that
>     optimal MSE distortion is attained by the sample-mean estimator $\frac{1}{P}\\sum_{i=1}^P \hat{X}\_i(y)$ as $P\rightarrow\infty$, while perfect perceptual quality is attained by $\hat{X}_i(y)$ for any $i$.
>     Thus, given $\\{\hat{X}\_i(y)\\}\_{i=1}^P$ with large $P$, there is essentially no PD tradeoff.
>
>     The construction of a *single* (possibly random) estimate $\hat{X}(y)$ from $y$ is  known as *point estimation*, even when $\hat{X}(y)$ is a random draw from the posterior. See
>     >   Lehmann and Casella, *Theory of Point Estimation*, Springer 2006.
>
>     Because the PD tradeoff assumes the use of a *single* estimate $\hat{X}(y)$ from $y$, it proves a property of point estimation.
>     We hope this clarifies our point.
>
>     That said, the PD tradeoff is non-essential to motivating pcaGAN, and removing it from the paper would in no way diminish our contribution.
>     Thus, **we plan to remove the PD discussion from our Introduction**.
>
> - **Reviewer**: I don't see why the rise of posterior sampling is linked to the PD tradeoff.
>
>     **Response**: See above.  However, since this point is non-essential to motivating pcaGAN, **we plan to remove the PD discussion from our Introduction**.
>
> - **Reviewer**: L30: "posterior samples will display attributes that reflect minority and majority populations in the training data". As far as I know, this is often incorrect.
>
>     **Response**: A proof of what we meant is given in App.A of [21].  However, since this statement is non-essential in motivating pcaGAN, **we plan to remove it**.
>
> - **Reviewer**: You are missing citations (and comparison) with highly related work proposed prior to rcGAN [21], for example [Ohayon'21] and [Man'24]. In fact, rcGAN [21] is based on [Ohayon'21] (regularize by enforcing agreement of the average posterior sample with the posterior mean) and on [Man'24] (use a second-moment penalty of the covariance matrix as well).
>
>     **Response**: We'd be happy to cite and compare to [Ohayon'21]; see the rebuttal pdf. As for [Man'24], it's highly specialized to the JPEG decoding application, and so it's not clear which modifications would be fair for MRI or inpainting.
>     We don't think that rcGAN is "based on" [Ohayon'21] or [Man'24], though.
>     The rcGAN paper [21] cites the 2017 Isola cGAN paper [17] for posterior-mean regularization.
>     And rcGAN predates [Man'24], as evident from the arXiv release dates.
>
> - **Reviewer**: I would suggest explaining L76-L79 better and including a self-contained background section
>
>     **Response**: Sure, we'd be happy to do so.
>
> - **Reviewer**: Some results seem wrong to me. For example, the results of DPS in Figure 4, which show inconsistencies between the outputs and the inputs. Also, I find it very hard to believe that a CGAN algorithm would outperform DPS when the hyper-parameters of DPS are properly tuned.
>
>     **Response**: Please see the global rebuttal and the rebuttal pdf.
>     Consistent with the reviewer's intuition, tuned DPS outperforms all existing cGANs under test.
>     However, it does not outperform the proposed pcaGAN, which attests to the importance of principal-component regularization.
>
> - **Reviewer**: The contribution of this work is quite limited in my opinion. 1) The proposed algorithm only slightly improves its predecessor, 2) A comparison with [Ohayon'21] and [Man'24] is missing, as well as recent works like DDNM and DDRM. 3) I would be happy if you could clarify the contribution of your work and its impact on the field, as it seems quite incremental to me.
>
>     **Response**: All three points are addressed in the global rebuttal.
>
> - **Reviewer**: I personally like this type of work and the proposed approach.
>
>     **Response**: We are glad to hear that you like our work.  We feel, however, that your final rating of "3" is at odds with this statement and with the remainder of your review.
>     For example, your first 3 critiques targeted non-essential statements in our Introduction.
>     Your 4th critique pertained to the missing references/comparisons to [Ohayon'21] and [Man'24], which are easily remedied.
>     Your 5th critique asked for more detail, which is easy to provide, and the remaining two critiques asked for comparisons to DDRM, DDNM, and tuned DPS, which we've included in the rebuttal.
>     With all this in mind, we kindly ask you to reconsider your rating.

---

> > ### Comment · Reviewer_cM4t · 2024-08-07
> >
> > I'd like to thank the authors for their thorough response.
> >
> > 1. I disagree regarding your explanation about the perception-distortion tradeoff. First of all, the paper does not only consider *single* draw estimators. In fact, the definition of distortion used in the paper is a mean taken over all Y's, and for each Y the mean is taken over all the random draws from the posterior and all the random draws from the estimator:
> > $$\mathbb{E}[\Delta(X,\hat{X})]=\int\int\int \Delta(x,\hat{x})p_{X|Y}(x|y)p_{\hat{X}|Y}(\hat{x}|y)p_{Y}(y)dxd\hat{x}dy$$
> > Namely, when talking about stochastic estimators (e.g., the posterior sampler), distortion (as defined in this paper) actually considers all possible draws from the estimator, for each Y. So, I still don't see why the perception-distortion tradeoff tells us anything "bad" about point estimators. Moreover, the perception-distortion tradeoff proves that the perception-distortion function,
> > $$P(D)=\min_{p_{\hat{X}|Y}}d(p_{X},p_{\hat{X}})\quad s.t.\quad \mathbb{E}[\Delta(X,\hat{X})]\leq D$$
> > is monotonically non-increasing in $D$, but it was never proven that this general function is *strictly* decreasing. For example, if we take the trivial distortion $\forall x,\hat{x}:\Delta(x,\hat{x})=0$, then $P(D)$ is constant in $D$ (it always equals zero). Thus, perception and distortion (as defined in the perception-distortion tradeoff paper) are not always at odds with each other. Lastly, I am sorry, but I don't understand what you mean by "with large $P$, there is essentially no PD tradeoff."
> >
> > 2. I don't see why would you remove the discussion about the perception-distortion tradeoff from your paper. It is central to motivating special types of regularizations when training with GANs/cGANS and not just a "regular" MSE loss. So actually, in my opinion, without discussing the perception-distortion tradeoff, the motivation of such work (e.g., pcaGAN) would be unclear.
> >
> > 3. Could you please point out to me where in the cGAN paper by Isola et al. "Image-to-Image Translation with Conditional Adversarial Networks" I can find the posterior mean regularization?
> >
> > Thank you!

---

> > > ### Author Response · Authors · 2024-08-08
> > >
> > > We thank the reviewer for the continued discussion and opportunity to improve our submission.
> > >
> > > 1. We appreciate the reviewer's invitation to further our discussion around the PD tradeoff and its connection to point estimates and posterior sampling. Our main point is that an estimator that returns $P\\gg 1$ posterior samples lets us trade perception with distortion in a way that a point estimator (which returns a single sample, such as a single posterior sample) cannot. Regardless of whether the reviewer agrees with us on the specifics of this point, we would like to emphasize that **this discussion is immaterial to the main ideas of our paper**, which revolve around enforcing cGAN consistency with the principal components of the true posterior covariance matrix. To avoid directly tying point estimation to PD or the motivation for posterior sampling to PD, we have revised our Introduction; please see our response to your next point.
> > >
> > > 2. We are removing the discussion of the PD tradeoff from our Introduction (not our entire paper) to definitively address the first and second critiques in your original review.  In particular, we propose to revise the second and third paragraphs of our Introduction as follows.
> > >
> > >     > In much of the literature, image recovery is posed as point estimation, where the goal is to return a single best estimate $\\hat{\\boldsymbol{x}}$. But there are shortcomings to this approach. For example, it’s not clear how to define “best,” since L2- or L1-minimizing $\\hat{\\boldsymbol{x}}$ are often regarded as too blurry, while perceptually pleasing $\\hat{\\boldsymbol{x}}$ may suffer from hallucinations [1–5]. Also, many applications demand not only a recovery $\\hat{\\boldsymbol{x}}$ but also some quantification of the uncertainty in that recovery [7,8].
> > >     >
> > >     > As an alternative to point estimation, posterior-sampling-based image recovery [10–28] has been proposed. There the goal is to generate $P$ samples $\\{\\hat{\\boldsymbol{x}}\_i\\}\_{i=1}^P$ from the posterior distribution $p\_{x|y}(·|\\boldsymbol{y})$. Posterior sampling facilitates numerous strategies to quantify the uncertainty in estimating $\\boldsymbol{x}$, or any function of $\\boldsymbol{x}$, from $\\boldsymbol{y}$ [7,8].
> > >     It also can help visualize uncertainty, and it can increase robustness to adversarial attacks [Ohayon,ICML'23].
> > >     That said, the design of accurate and computationally-efficient posterior samplers remains an open problem.
> > >
> > >     We would be interested to hear your feedback.
> > >
> > > 3. We should clarify what we wrote in our rebuttal.  The rcGAN paper [21] defines the "supervised-L1 loss" as $E\\{\\|\\boldsymbol{x}-\hat{\\boldsymbol{x}}\_{(P)}\\|\_1\\}$ with $\\boldsymbol{x}$ the true (training) image and $\hat{\\boldsymbol{x}}\_{(P)}$ the $P$-sample output average. On page 4, [21] says "regularizing a cGAN with supervised-L1 loss alone is not new; see, e.g., [Isola]".  But looking closely at [Isola], it seems that only the $P=1$ case is described there (i.e., no averaging), and so perhaps [21] is the first to propose this "supervised-L1 loss".  The paper [Ohayon'21] considers the squared-L2 version $E\\{\\|\\boldsymbol{x}-\hat{\\boldsymbol{x}}\_{(P)}\\|\_2^2\\}$, for which only the $P=1$ case is described in [Isola]. In any case, if accepted, our final paper will cite [Ohayon'21] for what appears to be the first cGAN regularization involving  $\hat{\\boldsymbol{x}}\_{(P)}$ with $P>1$.

---

> > > > ### Comment · Reviewer_cM4t · 2024-08-08
> > > >
> > > > 1. That sounds better and more accurate to me. There are still some things I would personally rephrase, which could be misleading to some readers. For example: "while perceptually pleasing $\hat{x}$ may suffer from hallucinations." The notion "hallucinations" carries a heavy load in several fields, so this could mislead readers about the meaning of producing perceptually pleasing $\hat{x}$. For example, researchers from the world of LLM would think that image reconstruction models that produce perceptually pleasing outputs often produce very wrong outputs, but any sample from the posterior distribution is actually not "wrong".
> > > >
> > > > 2. Note that you also need to revise L43 onwards, where you start talking about the posterior mean regularization.
> > > >
> > > > 3. In the rebuttal, you wrote: "We don't think that rcGAN is "based on" [Ohayon'21] or [Man'24], though. The rcGAN paper [21] cites the 2017 Isola cGAN paper [17] for posterior-mean regularization." But now you are saying that the Isola cGAN paper did not propose posterior-mean regularization? While your work may have good results, I think it is equally important to write a proper (and correct) introduction section where this work is properly positioned compared to previous/related literature.
> > > >
> > > > Could you please try to include here the entire new introduction so I could make sure it all makes sense to me?

---

> > > > > ### Author Response · Authors · 2024-08-09
> > > > >
> > > > > We thank the reviewer for the valuable feedback and continued discussion. Due to length constraints, we'll need to split our answer in two. Below is our proposed revision to the Introduction.
> > > > >
> > > > > > &nbsp;&nbsp;&nbsp;&nbsp;In image recovery, the goal is to recover the true image $\\boldsymbol{x}$ from noisy/distorted/incomplete measurements $\\boldsymbol{y} = \\mathcal{M}(\\boldsymbol{x})$.
> > > > > > This arises in linear inverse problems such as denoising, deblurring, inpainting, and magnetic resonance imaging (MRI), as well as in non-linear inverse problems like phase-retrieval and image-to-image translation. For all such problems, it is impossible to perfectly recover $\\boldsymbol{x}$ from $\\boldsymbol{y}$.
> > > > > >
> > > > > > &nbsp;&nbsp;&nbsp;&nbsp;In much of the literature, image recovery is posed as point estimation, where the goal is to return a single best estimate $\\hat{\\boldsymbol{x}}$. But there are shortcomings to this approach. For example, it’s not clear how to define “best,” since L2- or L1-minimizing $\\hat{\\boldsymbol{x}}$ are often regarded as too blurry, while efforts to make $\\hat{\\boldsymbol{x}}$ perceptually pleasing can sacrifice agreement with the true image $\\boldsymbol{x}$ and cause hallucinations [1–5], as we show later. Also, many applications demand not only a recovery $\\hat{\\boldsymbol{x}}$ but also some quantification of the uncertainty in that recovery [7,8].
> > > > > >
> > > > > > &nbsp;&nbsp;&nbsp;&nbsp;As an alternative to point estimation, there is posterior-sampling-based image recovery [10–28], which aims to generate $P$ samples $\\{\\hat{\\boldsymbol{x}}\_i\\}\_{i=1}^P$ from the posterior distribution $p\_{x|y}(\\cdot|\\boldsymbol{y})$. Posterior sampling facilitates numerous strategies to quantify the uncertainty in estimating $\\boldsymbol{x}$, or any function of $\\boldsymbol{x}$, from $\\boldsymbol{y}$ [7,8].
> > > > > > It also can help with visualizing uncertainty and increasing robustness to adversarial attacks [Ohayon,ICML'23].
> > > > > That said, the design of accurate and computationally-efficient posterior samplers remains an open problem.
> > > > > >
> > > > > > &nbsp;&nbsp;&nbsp;&nbsp;Given a training dataset of image/measurement pairs $\{(\\boldsymbol{x}\_t, \\boldsymbol{y}\_t
> > > > > )\}_{t=1}^T$, our goal is to build a generator $G\_{\\boldsymbol{\\theta}}$ that, for a given $\\boldsymbol{y}$, maps random code vectors $\\boldsymbol{z}\\sim\mathcal{N}(\\boldsymbol{0}, \\boldsymbol{I})$ to samples of the posterior, i.e., $\\hat{\\boldsymbol{x}} = G\_{\\boldsymbol{\\theta}}(\\boldsymbol{z}, \\boldsymbol{y}) \\sim p\_{x|y}(·|\\boldsymbol{y})$.
> > > > > > There are many ways to approximate the ideal generator.
> > > > > > The recent literature has focused on conditional variational autoencoders (cVAEs) [14–16], conditional normalizing flows (cNFs) [10–13], conditional generative adversarial networks (cGANs) [17–21,Ohayon'21,Man'24], and diffusion posterior samplers [22–28,Wang'23]. Although diffusion samplers have garnered a great share of recent attention, they tend to generate samples several orders-of-magnitude slower than their cNF, cVAE, and cGAN counterparts.
> > > > > >
> > > > > > &nbsp;&nbsp;&nbsp;&nbsp;With the goal of fast and accurate posterior sampling, recent progress in cGAN training has been made through regularization.
> > > > > > For example, Ohayon et al. [Ohayon'21] proposed to enforce correctness in the generated $\\boldsymbol{y}$-conditional mean using a novel form of L2 regularization.
> > > > > > Later, Bendel et al. [21] proposed to enforce correctness in the generated $\\boldsymbol{y}$-conditional mean *and* trace-covariance using L1 regularization plus a correctly weighted standard-deviation (SD) reward, and gave evidence that their "rcGAN" competes with contemporary diffusion samplers on MRI and inpainting tasks.
> > > > > > Soon after, Man et al. [Man'24] showed that L2 regularization and a variance reward are effective when training cGANs for JPEG decoding.
> > > > > > More details are given in Section 2.
> > > > > >
> > > > > > &nbsp;&nbsp;&nbsp;&nbsp;In a separate line of work, Nehme et al. [29] trained a Neural Posterior Principal Components (NPPC) network to directly estimate the eigenvectors and eigenvalues of the $\\boldsymbol{y}$-conditional covariance matrix, which allows powerful insights into the nature of uncertainty in an inverse problem.
> > > > > >
> > > > > > &nbsp;&nbsp;&nbsp;&nbsp;**Our contributions.**
> > > > > > Inspired by regularized cGANs and NPPC, we propose a novel “pcaGAN” that encourages correctness in the $K$ principal components of the $\\boldsymbol{y}$-conditional covariance matrix, as well as the $\\boldsymbol{y}$-conditional mean and trace-covariance, when sampling from the posterior.
> > > > > > We demonstrate that pcaGAN outperforms existing cGANs [18, 19, 21, Ohayon'21] and diffusion methods [24, 27, 28, Wang'23] on posterior sampling tasks like denoising, large-scale inpainting, and accelerated MRI recovery, while sampling orders-of-magnitude faster than [24, 27, 28, Wang'23].
> > > > > > We also demonstrate that pcaGAN recovers principal components more accurately than NPPC using approximately the same runtime.

---

> > > > > > ### Author Response · Authors · 2024-08-09
> > > > > >
> > > > > > [This is a continuation of our previous comment.]
> > > > > >
> > > > > > Regarding hallucinations, we do find that, relative to posterior-mean recoveries, posterior samples tend to hallucinate fine details that are not present in the true image, and we show evidence of that in Figures 3, C.1, E.4, E.5, E.6, and E.7. While a posterior sample as a whole may not be "wrong," it can fabricate fine details. Note that individual posterior samples are farther from the truth than the posterior mean recovery (in terms of average L2 distance) since they have 3dB worse MSE.
> > > > > >
> > > > > > In addition to revising our Introduction, we plan to add the following after L79 in Section 2:
> > > > > >
> > > > > > > We note that the $\\mathcal{L}\_{1,P}(\\theta)$ regularization proposed in [21] is closely related to the regularization $\\mathcal{L}\_{2,P}(\\theta) \\triangleq E\_{x,z1,...,zP,y}\\{\\|\\boldsymbol{x}-\\hat{\\boldsymbol{x}}\_{(P)}\\|_2^2\\}$ proposed earlier by Ohayon et al. in [Ohayon'21].
> > > > > > > A detailed discussion of the advantages and disadvantages of various cGAN regularizations can be found in [21].

---

> > > > > > > ### Comment · Reviewer_cM4t · 2024-08-11
> > > > > > >
> > > > > > > I sincerely thank the authors for the thorough discussion.
> > > > > > >
> > > > > > > I have given this a lot of thought. I reviewed the paper again and took the following into account:
> > > > > > >
> > > > > > > 1) Our discussion (including the revised introduction).
> > > > > > > 2) The additional experiments and results.
> > > > > > > 3) The reviews made by other reviewers, as well as the authors' responses.
> > > > > > >
> > > > > > > While I think the results are appealing, I believe the paper is not 100% ready for publication due to insufficient soundness and presentation (not only in the introduction). Therefore, I am:
> > > > > > >
> > > > > > > 1) Decreasing my scores for soundness and presentation.
> > > > > > > 2) Increasing my score for contribution.
> > > > > > > 3) Increasing my overall score to borderline reject (*given that the authors will incorporate the new introduction section*).
> > > > > > >
> > > > > > > Despite my opinion about the paper's soundness and presentation, I would like to emphasize that the paper's contribution is, in my opinion, worthy (given that all the results are 100% correct, which I have not confirmed). However, unfortunately, I can't confidently recommend acceptance for the current state of the paper.

---

### Official Review · Reviewer_GSQK · 2024-06-19

**Soundness:** 2
**Presentation:** 4
**Contribution:** 2
**Rating:** 5
**Confidence:** 4

**Summary:**

The authors address inverse problems and aim to improve GAN-based posterior samplers by adopting regularization to correct the principal components of the covariance matrix of the approximated posterior distribution.

**Strengths:**

1. The paper is well written and easy to follow.
2. The idea presented in the paper is novel and clear, especially given the tendency of GAN-based posterior samplers to experience mode collapse. This serves as a strong motivation for the proposed method in the paper.
3. The authors conducted empirical study with a variety of datasets.

**Weaknesses:**

1. For each dataset in the empirical study, a different set of baseline methods was chosen, possibly due to the requirement of pretraining. However, for a fair comparison, this is necessary for a complete research.
2. The authors claim that the proposed method outperforms existing diffusion models (line 52). However, the experiments conducted with diffusion models are not comparable due to the much smaller test size (line 210 in MRI experiments and line 235 in inpainting experiments).
3. Figure 4 presents inpainting posterior samples, but for the diffusion model referred to as DPS, the masking was not applied, which results in incorrect and misleading visualization.
4. Some minor errors:
    - In line 28, “There the goal" should be revised.
    - In lines 33-34, there is a grammar mistake.

**Questions:**

1. While diffusion models are slower compared to GANs, they have demonstrated success over GAN-based models in enhancing posterior richness for conditional tasks. A fair comparison with diffusion models would be valuable, as it remains uncertain whether the proposed method outperforms diffusion models. For faster sampling approaches, the authors might consider DDIM[1], which accelerates the diffusion process.

References:

[1] Song, Jiaming, Chenlin Meng, and Stefano Ermon. "Denoising diffusion implicit models." (2020).

**Limitations:**

The authors explained the limitations of their work.

---

> ### Author Rebuttal · Authors · 2024-08-06
>
> We sincerely thank the reviewer for their valuable feedback and time spent reviewing.
>
> - **Reviewer**: For each dataset in the empirical study, a different set of baseline methods was chosen.
>
>     **Response**: Yes, this was done because there exists no single baseline method that is state-of-the-art for all datasets considered.  Also, we wanted to compare against cGANs (because we propose a cGAN), diffusion-based methods (because they are recently popular), and NPPC (which offers a direct way to estimate posterior covariance principal components).  We don't think that using a strong, dataset-specific set of baselines would be a weakness.
>
> - **Reviewer**: The experiments conducted with diffusion models are not comparable due to the much smaller test size (line 210 in MRI experiments and line 235 in inpainting experiments).
>
>     **Response**: There must be a misunderstanding, since we always use a common test dataset when comparing any collection of methods.  For example, the CFID$^1$ and FID columns in Tab.2 evaluate all five methods on a common 72-image dataset.  Likewise, the CFID column in Tab.3 evaluates all four methods on a common 51k-sample dataset, and the FID column in Tab.3 evaluates all four methods on a common 6k-sample dataset.
>
>     Line 210 explains that, beyond the CFID$^1$ experiment that compared both cGANs and diffusion methods, we conducted *additional* cGAN-only experiments that used larger test datasets, which led to the CFID$^2$ and CFID$^3$ columns in Tab.2.  This was needed because Jalal's diffusion sampler is very slow to generate samples.  For example, evaluating Jalal for the CFID$^3$ columns in Tab.2 would require 72 days on our hardware.
>     Line 235 simply says that large datasets were used to avoid small-sample bias.
>
>
> - **Reviewer**: Figure 4 presents inpainting posterior samples, but for DPS, the masking was not applied, which results in incorrect and misleading visualization.
>
>     **Response**: Yes, our original submission used DPS under a default parameter setting, which was suboptimal for our experimental setup. The global rebuttal describes how DPS has since been tuned, and the rebuttal pdf shows the performance of tuned DPS.
>
> - **Reviewer**: A fair comparison with diffusion models would be valuable, as it remains uncertain whether the proposed method outperforms diffusion models. DDIM could be considered to accelerate sampling.
>
>     **Response**: Yes, we agree. The global rebuttal adds a comparison to DDRM and DDNM, which both use DDIM for acceleration.

---

> > ### Comment · Reviewer_GSQK · 2024-08-11
> >
> > I thank the author for addressing my concerns by evaluating several diffusion algorithms against their method. It appears that the DPS algorithm yields results nearly identical to the proposed approach, however, DPS heavily relies on a specific scale factor. Additionally, while the DPS algorithm is useful, it is not currently the SOTA for posterior sampling. Although I recognize the novel and valuable contribution of this idea for GANs, I am not convinced that it outperforms the performance of current diffusion models. Therefore, I remain my score as it is.

---

> > > ### Author Response · Authors · 2024-08-12
> > >
> > > We thank the reviewer for their valuable feedback and continued discussion.
> > >
> > > - **Reviewer**: I thank the author for addressing my concerns by evaluating several diffusion algorithms against their method. It appears that the DPS algorithm yields results nearly identical to the proposed approach, however, DPS heavily relies on a specific scale factor. Additionally, while the DPS algorithm is useful, it is not currently the SOTA for posterior sampling. Although I recognize the novel and valuable contribution of this idea for GANs, I am not convinced that it outperforms the performance of current diffusion models. Therefore, I remain my score as it is.
> > >
> > >     **Response**: Yes, our method and DPS had comparable performance after the latter was finely tuned for the FFHQ inpainting application. While we agree with the reviewer that diffusion models are powerful and hold significant promise for solving inverse problems, we emphasize that sampling speed is critical for many applications. Even with accelerated sampling methods, such as those based on DDIM, diffusion methods are still orders of magnitude slower than cGANs. For example, in most clinical setups, MR images are expected to be available immediately after the data acquisition. This level of latency cannot be achieved with current diffusion models. Therefore, posterior sampling methods that can compete with existing diffusion models while offering orders of magnitude faster image generation hold great practical value.

---

### Official Review · Reviewer_s7hc · 2024-07-10

**Soundness:** 4
**Presentation:** 4
**Contribution:** 2
**Rating:** 5
**Confidence:** 5

**Summary:**

The paper focuses on learning posterior samplers for inverse problems, specifically aiming to accurately estimate the principal components of the posterior covariance matrix. To achieve this, the authors consider training a conditional generative adversarial network (cGAN), building upon previous work [21]. They introduce two new regularization terms: one for the eigenvalues and another for the eigenvectors. Recognizing the challenge of estimating these statistics from a single ground truth solution per input measurement, the authors propose gradually adding these terms during training, while utilizing intermediate results to estimate the statistics.
The authors present a series of experiments to demonstrate their method, ranging from toy examples to real-world MRI data. Comparisons are primarily made with rcGAN and NPPC.

**Strengths:**

* The paper is well-written, with clear motivation, method presentation, and background (mostly).
* The detailed algorithm description, beyond pseudocode, significantly enhances understanding and reproducibility.
* The authors nicely addressed the problem of estimating the desired statistics from a single ground truth vector.
* The authors provide sufficient evidence to demonstrate their method (but additional evidence could further strengthen their claims).

**Weaknesses:**

Major Issues:
* The paper lacks an ablation study which is crucial. While the authors followed [21] and set $P_{rc}=2$, I still expect an ablation study on this parameter, especially since using 2 samples seems insufficient to estimate statistics properly.
Furthermore, there is no ablation study regarding the different regularization terms and their relative weights. Given the newly regularization terms, it is possible that other terms, introduced in [21], are redundant, especially when $K=d$. Since accurately recovering the eigenvalues implies perfect recovery of the trace, for example.

* I find the comparison to competing methods limited. I suggest adding comparisons to additional recent posterior sampling techniques. For example, the authors of NPPC presented comparison to DDRM, DDNM, RePaint, and MAT.

*  Unlike work [21], the paper present no proposition or theoretical guarantees regarding the estimated statistics, limiting the novelty of the paper.

* The description of work [21] is imprecise. Specifically, the usage of the notations $P$ and $P_{rc}$ is incorrect, leading to confusion.

* The authors state in lines 103-104 that they apply the regularization term (11) only after allowing the generated posterior mean to converge near the true mean. This raises the question of whether (11) is necessary or if its impact is marginal, as it only leads to small variations in the generated posterior mean. The same question applies to the generated eigenvalues. I suggest adding plots to illustrate the variation in generated statistics over training epochs, comparing them to their final values or their values before applying the new regularization term during training.

* While the relevance of NPPC to this work is clear, the experimental comparison seems unfair as NPPC is not designed for generation tasks. Furthermore, the authors demonstrate a speedup against DPS, but neglect to discuss the computational load of their method compared to NPPC.

Minor:
* The authors in lines 24-26 correctly point out that bias can be an issue in single-sample recovery. However, this is also a potential concern for recovery methods that yield multiple solutions.

**Questions:**

* Please address the weaknesses above.
* I suggest conducting an ablation study to assess the impact of key priors in the method, such as the different terms and $P_{rc}$.
* I suggest expanding the experiments to include comparisons with more recent posterior samplers in standard image inverse problems, like super-resolution and inpainting.
* Please provide plots illustrating the variation in the generated statistics over the training epochs.
* Do the additional regularization terms theoretically change the optimal solution, or does it remain the same?

**Limitations:**

See weaknesses.

---

> ### Author Rebuttal · Authors · 2024-08-06
>
> We sincerely thank the reviewer for their valuable feedback and time spent reviewing.
>
> - **Reviewer**: No ablation study of the rcGAN parameter $P_{rc}$, especially since $P_{rc}=2$ seems insufficient to estimate statistics properly.
>
>     **Response**: We use $P_{rc}=2$ when training an rcGAN model, which is the first step in training pcaGAN. To train rcGAN, the paper [21] analytically proved that $P_{rc}=2$ is sufficient and empirically found that $P_{rc}=2$ is optimal.
>     In the second and third stages of pcaGAN training, when the eigenvector and eigenvalue regularizations are introduced, we see no good reason to change $P_{rc}$ (nor any of the other rcGAN regularization parameters) because doing so would likely destabilize the training.
>     Given the clear rationale for $P_{rc}=2$, we feel that an ablation study is unnecessary, especially since it would take many weeks of compute.
>
> - **Reviewer**: The rcGAN posterior variance regularization is redundant when $K=d$.
>
>     **Response**: True, but pcaGAN is typically used with $K<d$, in which case variance regularization is essential.  Even when $K=d$, Fig.1b shows that there is no penalty to this redundancy, and Fig.1c shows that pcaGAN gives a significant improvement over both rcGAN and NPPC.
>
> - **Reviewer**: The comparison to competing methods is limited.  I suggest adding comparison to DDRM, DDNM, RePaint, and MAT.
>
>     **Response**: As described in the global rebuttal, have added results for DDRM and DDNM to the rebuttal pdf, and we have tuned DPS since the original submission.
>     We will include these results in our final paper if accepted.
>
> - **Reviewer**: No theoretical guarantees, limiting the novelty of the paper.
>
>     **Response**: pcaGAN's novelty is that it's the first posterior-sampling method to aim for correctness in the principal components of the posterior covariance matrix.  That novelty is independent of the presence/absence of theoretical guarantees.
>
> - **Reviewer**: The description of work [21] is imprecise. Specifically, the usage of the notations $P$ and $P_{rc}$ is incorrect.
>
>     **Response**: Definitions (7)-(9) are stated in terms of a generic variable $P$, and their usage in (6) correctly invokes $P=P_{rc}$ for rcGAN.  That said, we do see one typo in (7), where $P_{rc}$ should be replaced by $P$.  Is this what the reviewer was referring to?
>
> - **Reviewer**: There is a question of whether the eigenvector regularization (11) is necessary or if its impact is marginal. The same question applies to the eigenvalue regularization (12). I suggest adding plots to illustrate the variation in generated statistics over training epochs.
>
>     **Response**: There must be a misunderstanding here.  The regularizations in (11)-(12) are what distinguishes the proposed pcaGAN from the existing rcGAN.  To understand the impact of (11)-(12) on performance, one needs only to compare the rcGAN and pcaGAN entries in our figures and tables.
>     A summary of the performance gains is given in the global rebuttal.
>
> - **Reviewer**: The experimental comparison to NPPC seems unfair as NPPC is not designed for generation tasks.
>
>     **Response**: There must be a misunderstanding here, since we evaluate NPPC only for non-generative tasks.  Recall that NPPC aims to directly estimate the principal components of the posterior covariance matrix, and that the NPPC paper [29] evaluated performance using Wasserstein-2 distance, residual MSE, and residual error magnitude.  We compare to NPPC using exactly these metrics, which is fair.
>
> - **Reviewer**: The authors demonstrate a speedup against DPS, but not NPPC.
>
>     **Response**: That's a good point.  For a batch of 128 MNIST samples, NPPC's inference time is 112 ms while pcaGAN's is 118 ms, a 5\% increase.  These results will be added to Tab.1.
>
> - **Reviewer**: Do the added regularization terms theoretically change the optimal solution, or does it remain the same?
>
>     **Response**: The added regularization terms do not change the optimal solution.  Recall that the pcaGAN regularizations aim to make the cGAN's $y$-conditional mean, trace-covariance, covariance eigenvectors, and covariance eigenvalues all agree with the respective statistics of the *true* posterior distribution.  So if the cGAN's output statistics are optimal, then the regularizations will have no effect.  But if the cGAN's $y$-conditional statistics are suboptimal, the regularizations will try to to correct them.

---

> > ### Comment · Reviewer_s7hc · 2024-08-11
> > **Post-Rebuttal**
> >
> > I thank the authors for their rebuttal and clarifications, which have been very helpful in addressing most of my concerns.  However, to further strengthen the paper, I still encourage an ablation study on the $P_{rc}$  parameter, or at least a more detailed discussion of its applicability to the proposed method, as the analysis in [21] may not directly transfer.
> >
> > While I acknowledge the novelty of pcaGAN in aiming for better posterior sampling, I believe theoretical guarantees would make the contribution even more impactful, similar to rcGAN. Additionally, an ablation study on the different regularization terms is recommended, as some original rcGAN terms might be redundant with the new ones and could potentially be removed to facilitate the training process.
> > Currently, I believe my score accurately reflects the merits of the work.

---

### Official Review · Reviewer_n2zA · 2024-07-12

**Soundness:** 3
**Presentation:** 4
**Contribution:** 3
**Rating:** 6
**Confidence:** 4

**Summary:**

This work considers using posterior sampling to solve inverse problems via conditional GANs. Previous work has shown that cGANs are competitive in terms of reconstruction quality as compared to other generative models to sample from the posterior (e.g., diffusion models), while also having the advantage that, once trained, sampling is fast as it simply requires a forward pass. In this paper, the authors build on such approaches by adding regularization terms to encourage the cGAN to learn K principal components of the posterior covariance matrix, where K is user-defined quantity. Due to the computational burden of such an approach, a training scheme is devised where learning of the eigenvectors and eigenvalues is done after optimizing a regularized cGAN objective. The method is validated on a variety of linear inverse problems (denoising, accelerated MRI, inpainting) and datasets (mnist, fastmri, ffhq).

**Strengths:**

- The paper is well-written and easy to read. In particular, the idea of adding regularization to fit to principal components of the posterior distribution is a simple and natural idea. It is also one of importance, as being able to visualize uncertainty in reconstructions is crucial in understanding which regions of the reconstruction can be “trusted”
- The method is validated on a nice range of problem types and datasets.
- Using a conditional cGAN allows for fast sample generation, especially as compared to diffusion models.
- The additional use of the pca regularization term appears to improve performance of rcGAN, while also allowing for visualization of directions of uncertainty.

**Weaknesses:**

- The proposed approach is a slight, potentially somewhat incremental, improvement to the prior rcGAN approach of [21]. In particular, rcGAN also aims to better model the posterior distribution by matching certain statistics of the posterior mean and covariance. This work takes this reasoning a step further by aiming to fit the principle components of the covariance. This has also been done (in a different way) in prior work [29], that the authors compare to. The bonus for this work, however, is that the authors can sample quickly, whereas in [29] sampling is not possible.
- Coupled with the concern regarding novelty, the additional regularization terms are computationally expensive to optimize, especially for high-dimensional problems. This is evidenced by, for example, only being able to set K = 1 in the MRI experiments and K = 2 in the large-scale inpainting experiments, and is discussed by the authors in their limitations. Moreover, the training strategy becomes a bit more complicated in order to aid in efficiency, as one needs to first optimize the rcGAN objective and then integrate the PCA terms into the learning process.

**Questions:**

- As the parameter $K$ is fixed prior to training with knowledge of the inverse problem, it would be good to discuss how the choice of $K$ relates to the particular inverse problem itself. For example, how does the choice of $K$ relate to how challenging the inverse problem is? I presume that ill-posedness plays a role here (e.g., comparing a linear inverse problem with a non-linear inverse problem, such as phase retrieval).

**Limitations:**

The authors discuss some of the above weaknesses I mentioned near the end of their paper, which I appreciate. They also discuss other issues regarding how to properly incorporate this approach for rigorous uncertainty quantification.

---

> ### Author Rebuttal · Authors · 2024-08-06
>
> We sincerely thank the reviewer for their valuable feedback and time spent reviewing.
>
> - **Reviewer**: Slight, potentially somewhat incremental, improvement to the prior rcGAN approach of [21].
>
>     **Response**: Please see the global rebuttal.
>
> - **Reviewer**: Additional regularization terms are computationally expensive to optimize in high-dimensional problems, forcing $K=1$ for multicoil MRI and $K=2$ for 256x256 image inpainting.
>
>     **Response**: True, our limited GPU memory forced us to use small $K$.
>     But even with $K=1$ and $K=2$, pcaGAN achieved noticeable performance gains over rcGAN (see global rebuttal).
>     And, with the memory advances predicted by Moore's law, pcaGAN's advantage will grow.
>     Regarding time complexity, pcaGAN is only 1.3x more expensive than rcGAN in training, and equal to rcGAN in inference.
>
> - **Reviewer**: How does the choice of $K$ relate to the inverse problem?
>
>     **Response**: Since $K$ is the number of posterior covariance principal components that pcaGAN aims to match, the effect of $K$ in pcaGAN is similar in spirit to the effect of $K$ in standard PCA. For example, pcaGAN's recovery performance should be non-decreasing in $K$, as corroborated by Fig.1b.  But when the singular values of the posterior covariance matrix decay quickly, a small $K$ will suffice for good recovery performance.

---

> > ### Comment · Reviewer_n2zA · 2024-08-09
> >
> > Thank you to the authors for their detailed rebuttal and for responding to my comments. I appreciate that it is still a non-trivial task to extend rcGAN to fit to the principal components and singular values and there are certainly empirical benefits to the proposed approach. I was hoping to ask further about the choice of $K$.
> >
> > It is mentioned that the choice of $K$ behaves similarly to PCA on the posterior covariance, which makes sense. But I think I was asking more specifically about if there is insight from the form of the inverse problem $y = \mathcal{M}(x)$ that can give insight into the conditioning of the posterior covariance. For example, do ill-conditioned linear forward models induce posterior covariance matrices that have slower decaying singular values? Similarly, if the inverse problem is non-linear, inducing a multi-modal posterior, is it necessarily the case that we should expect the posterior covariance to also have slower decaying singular values (as compared to a less, multi-modal posterior)? I ask these questions from the perspective of giving the reader some insight into this choice of $K$.

---

> > > ### Author Response · Authors · 2024-08-09
> > >
> > > Thanks, this is a great question. In general, the structure of the posterior covariance matrix will be governed by the interaction between the likelihood (defined by $\\boldsymbol{y}=\\mathcal{M}(\\boldsymbol{x})$ if we think of $\\mathcal{M}(\\cdot)$ as stochastic) and the prior on $\\boldsymbol{x}$.  One can construct toy problems where the posterior covariance has an intuitive structure, but generally speaking the likelihood, prior, and their interaction will be complicated and non-intuitive.
> > >
> > > As a toy example, consider a noiseless inpainting problem, where the first half of the entries in $\\boldsymbol{x}$ are directly observed.  Then the posterior distribution over those first-half entries would be a Dirac delta, while the posterior distribution over the last-half entries would be the *prior* distribution of the last-half entries conditioned on the first-half entries.
> > > For some priors, knowing the first-half entries tells you everything about the last-half entries, while for other priors it tells you nothing, and for most cases it gives you partial knowledge.
> > > So, little can be said without a good understanding the prior and, in particular, its interaction with the likelihood.
> > >
> > > As far as multi-modality is concerned, it does not directly impact the covariance structure, since covariance is based only on second-order statistics.  Again, consider a noiseless inpainting problem.  If the prior is a high-variance white Gaussian, then the posterior will be a high-variance white Gaussian in the unobserved dimensions.  If the prior is instead a Gaussian-mixture with many randomly located component means, the posterior will also be a Gaussian mixture and thus will generally be multi-modal in the unobserved dimensions.  But the posterior *covariance* will just measure the uncertainty *energy* along different dimensions, which might be very similar in the uni-modal and multi-modal cases.

---

> > > > ### Comment · Reviewer_n2zA · 2024-08-12
> > > >
> > > > Thank you for delving deeper into this. Indeed, it seems like there is not a direct mapping between the inverse problem structure and the posterior covariance, as the (unknown) prior plays a heavy role in determining the structure of the posterior. After carefully considering the author's comments and the other reviewer's discussions, I believe that my score still accurately reflects my view of the paper. Hence I will keep my score.

---

### Author Rebuttal · Authors · 2024-08-06

We sincerely thank all the reviewers for their valuable feedback and time spent reviewing.
In this global rebuttal, we address remarks made by multiple reviewers.

- **Reviewers n2zA and cM4t**: Incremental contribution over the prior work rcGAN [21]?

    **Response**: No.
    rcGAN enforces accuracy in the posterior mean and trace-covariance.
    The proposed pcaGAN goes beyond that by also forcing accuracy in the principal components of the posterior covariance matrix.
    To our knowledge, pcaGAN is the first-ever generative method to do so.
    Accomplishing this task was non-trivial, as it required novel eigenvalue and eigenvalue regularizations, as well as a novel training protocol to properly integrate them with the posterior mean and trace-covariance regularizations.

    There are two main advantages to pcaGAN over rcGAN.
    First, pcaGAN allows one to ascertain the *dominant modes of reconstruction uncertainty*.
    Although this is also attempted by NPPC [34], the Gaussian and MNIST results in Figs.1-2 and Tab.1 show that the eigenvectors and eigenvalues estimated by pcaGAN are significantly more accurate.

    Second, pcaGAN generates significantly more accurate posterior samples than rcGAN.
    Summarizing the results from our experiments, we find the following improvements from rcGAN to pcaGAN:

    - 1e-2/1e-3=10x improvement in $\\mathcal{W}_2$ distance in Fig. 1

    - 1.69/1.29=1.3x or 30\% improvement in CFID$^3$ in Tab.2 at $R=4$

    - 3.83/3.21=1.16x or 16\% improvement in CFID$^3$ in Tab.2 at $R=8$

    - 0.0379/0.0344=1.1x or 10\% improvement in LPIPS in Tab.3

    - 0.0877/0.0799=1.1x or 10\% improvement in DISTS in Tab.3

    - 3.74/3.45=1.08 or 8\% improvement in CFID in Tab.4

    Finally, Tab.A of the rebuttal pdf shows that *pcaGAN outperformed DPS in all three quality metrics (CFID, FID, and LPIPS), whereas rcGAN did not outperform DPS in any quality metrics*.  Thus pcaGAN is a significant improvement over rcGAN.

- **Reviewers GSQK and cM4t**: The DPS images do not look right.  Was DPS properly tuned?

    **Response**: In our original submission, we set the DPS step-size $\zeta'=1.0$, as suggested by the DPS paper [28] for FFHQ inpainting. But, with this value of $\zeta'$, the DPS inpainting recoveries were inconsistent with the original images at the unmasked pixels.  Thus, when computing the CFID and FID values in Tab.4, we replaced the DPS-recovered pixels with the true ones at the unmasked locations, which greatly improved the CFID and FID values reported for DPS.

    Just after the NeurIPS deadline, we discovered that DPS can be improved by tuning $\zeta'$ to our experimental setup.
    In particular, we found that, over the grid [0.5: 0.25: 3.0], the value $\zeta'=2.0$ minimizes LPIPS on a 1k-sample validation dataset.
    We used LPIPS because, as a feature-based metric, it correlates well with CFID and FID.
    (Note that CFID and FID themselves are inadmissible for tuning, because their values are unreliable with 1k-sample validation [36].)

    In the rebuttal pdf, Fig.A shows tuned-DPS inpainting recoveries, which look as expected.
    Also, Tab.A reports the performance of tuned-DPS on a 20k-sample test set.
    There we see that tuned-DPS outperforms all tested diffusion methods and cGANs in CFID, FID, and LPIPS *except* for pcaGAN.


- **Reviewers s7hc, GSQK, and cM4t**: How does pcaGAN compare to recent diffusion posterior samplers like DDRM, DDNM, or RePaint?

    **Response**: For multicoil fastMRI brain data, Jalal [24] is, to our knowledge, the state-of-the-art diffusion posterior sampler. We are unaware of any pretrained diffusion denoisers that can be used with samplers like DPS, DDRM, or DDNM to yield competitive performance on this dataset.
    We tried training our own diffusion model on the multicoil fastMRI brain data, but the results were not competitive.

    For FFHQ face data, however, there are readily available pretrained diffusion models that are compatible with DPS, DDRM, DDNM, and RePaint.
    Thus, in the rebuttal pdf, we add new results for DDNM (ICLR'23) and DDRM (NeurIPS'22).
    (We did not test RePaint, because DDNM has been shown to outperform it.)
    Fig.A shows example recoveries for pcaGAN, DPS (@ 1000 NFEs), DDNM (@ 100 NFEs), and DDRM (@ 20 NFEs).
    Tab.A shows CFID, FID, LPIPS, and the time to generate 40 samples.
    As expected from their NFEs, when going from DPS to DDNM to DDRM, we see an increase in recovery speed but a decrease in recovery quality.
    Notably, *pcaGAN outperforms all diffusion methods in both recovery quality and speed*.

    That said, comparisons between cGANs and diffusion methods should be taken with a grain of salt.
    cGANs are trained to handle a specific inverse problem, whereas diffusion samplers must handle a wide range of problems without retraining.
    Thus it is not surprising that a well-designed cGAN can outperform today's diffusion samplers.
    On the other hand, because cGANs use one NFE but diffusion samplers use many, it would not be surprising if tomorrow's diffusion samplers outperformed pcaGAN.


- **Implementation details for the new results in the rebuttal pdf**:
    - For DDNM, we use the authors' implementation and the suggested DDIM setup of 100 steps with $\eta=0.85$.
    - For DDRM, we use the authors' implementation and the suggested DDIM setup of 20 steps with $\eta=0.85$. Per the authors' suggestion, we also set $\eta_b=1$.
    - DDRM, DDNM, and DPS all use the same FFHQ diffusion model.
    - For pscGAN, we use the authors' suggestion to set $\lambda_{MM}=0.001$ and $M=8$ during training. To ensure a fair comparison, other components of the training/validation/test process are identical across all cGANs.

---

### Decision · Program_Chairs · 2024-09-25

**Decision:**

Accept (poster)

**Comment:**

In this paper, the authors present a method for using a conditional GAN to solve inverse problems by posterior sampling.  Whereas prior work enforces accuracy in the posterior mean and trace-covariance, this work further enforces accuracy in a user-specified number of principal components of the posterior covariance matrix.  The proposed work yields more accurate posterior samples than baselines with only modest increases in computational cost during training (relative to the most similar method to the proposed work).  Based on reviewer comments, there is some room for improvement in terms of thorough ablations and explanation of prior literature.  Based on the author-reviewer discussions, these concerns have been resolved enough to warrant acceptance.  The authors should implement the promised changes in the camera-ready.